METHODS AND RESOURCES

# The COMBO window: A chronic cranial implant for multiscale circuit interrogation in mice

Bradley J. Edelman[1,2,3,4☯]*, Dominique Siegenthaler[1,3,4☯], Paulina Wanken[1,3,4], Bethan Jenkins[4,5,6], Bianca Schmid[2], Andrea Ressle[2], Nadine Gogolla[2], Thomas Frank[4,5,6], Emilie Macé[1,3,4]*

**1** Brain-Wide Circuits for Behavior Research Group, Max Planck Institute for Biological Intelligence, Planegg, Germany, **2** Emotion Research Department, Max Planck Institute of Psychiatry, Munich, Germany, **3** Dynamics of Excitable Cell Networks Research Group, Department of Ophthalmology, University Medical Center Göttingen, Göttingen, Germany, **4** Cluster of Excellence "Multiscale Bioimaging: from Molecular Machines to Networks of Excitable Cells" (MBExC), University of Göttingen, Göttingen, Germany, **5** Olfactory Memory Research Group, Max Planck Institute for Biological Intelligence, Planegg, Germany, **6** Olfactory Memory and Behavior Research Group, European Neuroscience Institute and Faculty for Biology and Psychology, University of Göttingen, Göttingen, Germany

☯ These authors contributed equally to this work.
* bradley_edelman@psych.mpg.de (BJE); emilie.mace@med.uni-goettingen.de (EM)

**Data Availability Statement:** All data needed to evaluate the conclusions in the paper are present in the paper or the Supplementary Materials. All relevant COMBO window files, supplementary data

## Abstract

Neuroscientists studying the neural correlates of mouse behavior often lack access to the brain-wide activity patterns elicited during a specific task of interest. Fortunately, large-scale imaging is becoming increasingly accessible thanks to modalities such as $Ca^{2+}$ imaging and functional ultrasound (fUS). However, these and other techniques often involve challenging cranial window procedures and are difficult to combine with other neuroscience tools. We address this need with an open-source 3D-printable cranial implant—the COMBO (**Ch**r**O**nic **M**ultimodal imaging and **B**ehavioral **O**bservation) window. The COMBO window enables chronic imaging of large portions of the brain in head-fixed mice while preserving orofacial movements. We validate the COMBO window stability using both brain-wide fUS and multi-site two-photon imaging. Moreover, we demonstrate how the COMBO window facilitates the combination of optogenetics, fUS, and electrophysiology in the same animals to study the effects of circuit perturbations at both the brain-wide and single-neuron level. Overall, the COMBO window provides a versatile solution for performing multimodal brain recordings in head-fixed mice.

## Introduction

Acquiring large neural activity datasets from distant and interconnected regions is paramount to understanding the neural correlates of behavior [1,2]. For animal models with small brains, such as zebrafish and fruit flies, optical imaging approaches can fulfill this need by acquiring whole-brain activity during behavior at cellular resolution [3,4]. By contrast, in the larger

and plotting codes are freely available at the Zenodo repository https://doi.org/10.5281/zenodo.11092491. All raw fUS and two-photon data, along with the corresponding videographic recordings are freely available at the Edmond Data Repository https://doi.org/10.17617/3.5KYYHK.

**Funding:** This work was funded by the Max-Planck Society and the Deutsche Forschungsgemeinschaft (DFG, German Research Foundation) under Germany's Excellence Strategy - EXC 2067/1-390729940). B.J.E. was supported by the European Molecular Biology Organization Postdoctoral Fellowship no. ALTF 449-2020. D.S. was supported by the Swiss National Science Foundation (SNSF) Early Postdoc.Mobility no. 194957, SNSF Postdoc.Mobility no. 211087 and Deutsche Forschungsgemeinschaft Walter-Benjamin Programm (Stelle) no. SI 2831/1-1. The funders had no role in study design, data collection and analysis, decision to publish, or preparation of the manuscript.

**Competing interests:** The authors have declared that no competing interests exist.

**Abbreviations:** ACC, anterior cingulate cortex; COMBO, ChrOnic Multimodal imaging and Behavioral Observation; DMS, dorsal-medial striatum; fMRI, functional magnetic resonance imaging; fUS, functional ultrasound; GFAP, glial fibrillary acidic protein; GLM, general linear model; HOG, histogram of oriented gradients; LGN, lateral geniculate nucleus; MR, magnetic resonance; PBS, phosphate-buffered saline; PC, principal component; PMP, polymethylpentene; ROI, region-of-interest; SC, superior colliculus.

mouse brain, no such optimal technique is currently available. Large-scale recording modalities such as functional magnetic resonance imaging (fMRI), wide-field $Ca^{2+}$ imaging, and more recently functional ultrasound (fUS) imaging exhibit a relatively low spatiotemporal resolution. On the other hand, techniques with cellular resolution such as two-photon microscopy and electrophysiology suffer from limited brain coverage. Therefore, as no single modality is capable of imaging whole-brain activity in mice at the level of individual neurons, alternative strategies using existing tools must be employed. One recent strategy is to utilize techniques with low brain coverage (e.g., neuropixel recordings) in a highly parallelized manner to acquire large quantities of neuronal recordings across different regions of the brain during the same standardized behavioral task [5–7]. An alternative strategy that is more suitable to non-standardized behaviors is to combine different modalities such that information from multiple spatiotemporal scales is acquired in the same animals. In this case, large-scale recordings can inform which individual regions should be investigated in more detail without a priori knowledge [8]. Beyond acquiring neural data, it is increasingly popular to manipulate specific neural circuits (e.g., with optogenetics) to determine their causal impact on behavior. Combining optogenetics and large-scale imaging is a powerful approach to identify unexpected regions modulated by specific manipulations and to guide targeted recordings [9]. However, combining all these techniques in a single animal is technically challenging and therefore relatively rare.

A common feature of some of the most popular techniques used in neuroscience (electrophysiology, wide-field imaging, two-photon microscopy, optogenetics, fUS) is that they are most often applied through a cranial window. This is either because direct access to the brain is needed, e.g., for electrode or fiber implantation (electrophysiology, optogenetics, fiber photometry), or to maximize imaging quality and depth (optical imaging, fUS) [10–12]. Early chronic cranial windows developed specifically for optical imaging reported clear optical access to the brain at cellular resolution for months at a time [13,14]. However, these windows utilize flat glass coverslips that can cause anatomical distortions when placed over large areas of curved tissue. Therefore, such approaches are limited in spatial extent, often covering only 2 to 5 mm$^2$ of the brain [15]. A solution to this problem was provided by the "Crystal Skull" implant, a curved glass coverslip giving access to 75 mm$^2$ of the dorsal cortex and which is commonly used for wide-field imaging [16]. However, glass windows are not compatible with methods such as electrophysiology and fUS as glass strongly attenuates ultrasound waves and is difficult to penetrate with a probe. Recently, this restriction has been largely overcome with other geometry-based approaches that involve shaping a plastic film to the curvature of the skull and brain [17,18]. Such examples are also accompanied by curved implant frames that can be easily attached to the skull and combined with a lightweight head plate for head-fixation under an imaging apparatus [17,18]. In particular, the "See-Shell" implant designed by Ghanbari and colleagues [18] provides access to 45 mm$^2$ of the dorsal cortical surface and can be 3D-printed/laser cut using low-cost machines often found in laboratory settings. Such window designs exhibit robust long-term functionality, minimize the impact on brain tissue, and suggest a standardized surgical procedure. Despite these advantages, such designs have not been tested for acoustic imaging and are still limited in field of view. In parallel, other implants have been proposed to specifically accommodate fUS while being compatible with optical methods [19–22]. However, these implants are designed primarily for single-slice fUS acquisitions and therefore exhibit limited coverage, whereas volumetric fUS can now acquire a much larger portion of the mouse brain in a single acquisition (approximately 1 cm$^3$) [23]. Finally, the impact of available chronic cranial window implants on mouse behavior has been relatively unexplored even though orofacial movements are increasingly popular readouts of arousal, whisking, sniffing, or emotional states in head-fixed contexts [24,25].

Here, we propose to simplify multimodal imaging across spatial-temporal scales in head-fixed behaving mice using a chronic cranial implant, termed the **ChrOnic Multimodal imaging and Behavioral Observation (COMBO) window**. The COMBO window aims to combine and expand the advantages of currently available cranial window implants by providing (1) a larger field of view; (2) compatibility with optical and acoustic imaging; (3) integration with local recording and manipulation methods; and (4) unobstructed access to head-fixed behavioral readouts. We first validated modality-specific versions of the COMBO window for awake imaging with both fUS and two-photon $Ca^{2+}$ imaging. We then further validated an additional variant of the COMBO window with the combination of fUS, optogenetics, behavioral analyses, and electrophysiology in the same animals. To increase accessibility, we developed all components of the COMBO window and head fixation components for 3D printer or laser cutter production and provide all relevant files (**S1**–**S16 Files, https://doi.org/10.5281/zenodo.11092491**). By doing so, the most appropriate off-the-shelf option can be used for individual applications and can be 3D-printed in-house without the need for significant customization. Overall, the COMBO window provides a unified solution to the requirements of a large variety of neuroscientific experiments that benefit from access to a large portion of the mouse brain during behavioral tasks.

## Results

### Design principles of the COMBO window

The COMBO window consists of 3 parts: a resin-based implant frame, a metallic head plate, and a protective transparent film (**Figs 1A and S1**). We chose an asymmetric design to facilitate unobstructed videography of the animals' full face and body during head-fixed experiments. To achieve high stability and durability of the COMBO window, we fit the core structure closely to the surface of a standard mouse skull [18] (**Fig 1A**). We have successfully tested the COMBO window on both C57BL/6 and CH3 mice. As skull morphology is similar across mouse strains [26], we expect no issues also installing the COMBO window on other mouse lines. The COMBO window harbors a central hole providing access to an area of approximately 90 mm$^2$ of the dorsal skull, spanning from just posterior to the olfactory bulbs to the cerebellum in the anterior-posterior axis (**Fig 1B and 1C**). Thus, the majority of the cerebral cortex, striatum, pallidum, hippocampus, thalamus, hypothalamus, and midbrain can be accessed through this opening (**Fig 1C**). Flat protrusions with screw holes extending from the side and rear of the COMBO window, as well as a notch at the front, act as attachment points for the head plate. Furthermore, to enable the widespread use of the COMBO window across labs and to provide a unified solution for multiple experimental needs, we provide numerous versions (**Fig 1D**) that are all compatible with our custom-designed head plate and head plate holder (**Fig 1E**). First, to facilitate whole-brain fUS recordings, we added an upward protruding ring from the surface of the implant frame to contain the ultrasound gel during extended imaging sessions. For this reason, we termed this the "cup" version and it is intended for fUS applications. Second, to make the COMBO window compatible with optical modalities such as two-photon $Ca^{2+}$ imaging, we created a "flat" version without the cup to accommodate the limited working distance of typical high numerical aperture objectives used in two-photon imaging. Note that fUS can also be performed through the flat version and that only the external rim differs between the 2 versions (i.e., the film itself is not flat, as explained below). Third, to complement these 2 standard versions, we designed additional variations compatible with circuit manipulation techniques that require a physically implanted object such as an optical fiber or electrode (**Fig 1D**). As recording and stimulation sites vary depending on the scientific question, we created seven variations with different implantation compartments that

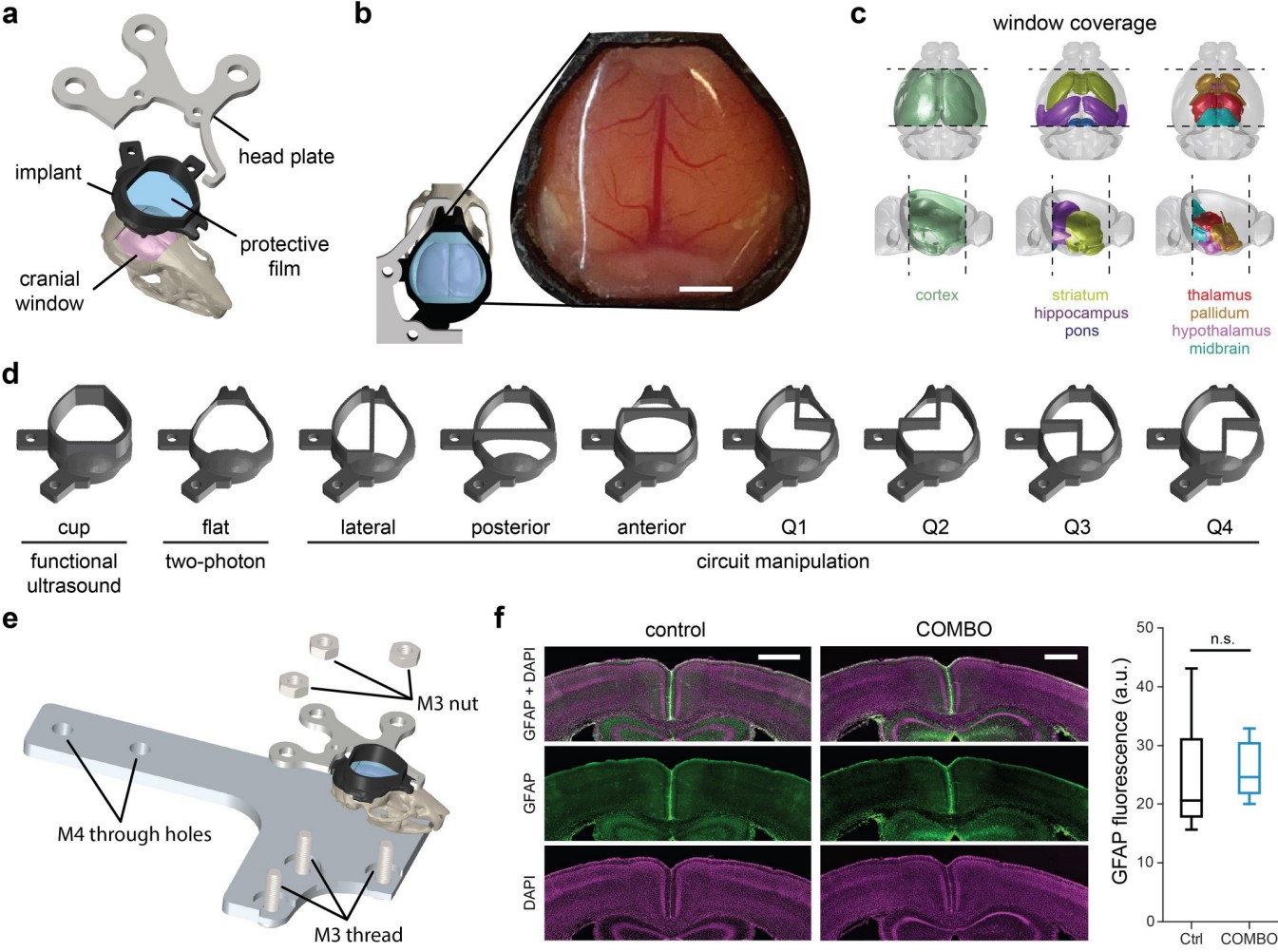

**Fig 1. Design principles of the COMBO window.** (**a**) CAD of the COMBO window and corresponding head plate. The core shape of the implant was fitted to the surface of a standard mouse skull, sealed with a protective film, and accompanied by a standard head plate. (**b**) Top-view of the implant CAD design. An example image of a whole-brain cranial window and the COMBO window installed on a mouse 4 weeks after surgery. Scale bar represents 2 mm. (**c**) 3D diagrams displaying the estimated window coverage, which includes the cerebral cortex, striatum, hippocampus, thalamus, hypothalamus, pallidum, midbrain, and pons. (**d**) Two whole-brain implant designs optimized for functional ultrasound imaging ("cup") and two-photon fluorescence microscopy ("flat"). Seven additional designs (with a cup) were created to accommodate a chronic implant (fibers, electrophysiology probes, etc.). (**e**) CAD design of the standard head plate holder that provides stable head fixation for the COMBO window. (**f**) GFAP fluorescence in the cortex (bregma −1.0 mm AP) of a representative control animal (left) and one with (right) the COMBO window installed. Images show the GFAP + DAPI (top), GFAP (middle) and DAPI (bottom) signal. Scale bars represent 500 μm. There was no significant difference in cortex-wide GFAP immunofluorescence between animals without ($N = 5$ mice, black) and with the COMBO window ($N = 6$ mice, blue). Boxplots represent the median (center line), 25th and 75th percentiles (lower and upper box), and the first and 99th percentile (whiskers). One-way repeated measures ANOVA on ranks. Main effect of the COMBO window: n.s. $p > 0.05$. Underlying data can be found in **S1 Data** and code in **S1 Code**. CAD, computer-aided design; COMBO, ChrOnic Multimodal imaging and Behavioral Observation; GFAP, glial fibrillary acidic protein.

accommodate different target regions without the need for additional modifications. While these compartments reduce the maximum cranial window size, such variations still provide access to an opening of between 50 and 80 mm$^2$ of the dorsal skull. Finally, we also provide a separate version that is compatible with magnetic resonance (MR) receive coils (**S2 Fig**). With this version, animals can be scanned with MRI before attaching the metallic head plate using a standard ear bar head fixation. While we did not extensively evaluate this version for potential MR artifacts in the current work, previous studies with implants of a similar material did not significantly distort MR images [27]. Therefore, we expect a similar result for the COMBO

window. 3D-printable files (.stl) for all versions of the COMBO window are provided in **S1**–**S10 Files**, a laser-cutter file for the universal head plate (.dwg) in **S11 File**, and various files for the head plate holder in **S12**–**S15 Files**. All files are additionally provided at an open-source repository for free public download (**https://doi.org/10.5281/zenodo.11092491**).

The COMBO window is sealed with a 125-μm thick polymethylpentene (PMP) film, which is slightly arched using a 3D-printed mold (**S16 File**) that mimics the curvature of the skull (**S3A Fig**). This ensures that the distance between the brain surface and the film is constant across the entire cranial window. By testing different thicknesses, we have found that 125-μm PMP offers an optimal compromise specifically for the COMBO window between flexibility and durability. PMP was chosen for its good acoustic properties; however, it is likely that others polymers (e.g., PET, as used with the See-Shell [18]) are also compatible with fUS imaging [11,18]. We provide detailed schematics and instructions for the preparation and installation of the COMBO window, as well as for the cranial window surgery itself (**S1 Appendix** and **S3 Fig**), to encourage optimal and consistent use. The overall success rate of COMBO window installation across users in our lab was 82% (4 surgeons, 53/65 mice). The success of this procedure is based on strict animal welfare criteria in the post-operative phase. Failure points involved welfare endpoints being reached during the recovery period or loss of the implant due to insufficient dental cement adhesion; such cases typically occurred within 1 week of surgery. A subset of the mice with successful COMBO window installation was consistently scanned with fUS for 8 weeks or more. In only 11% (4/37) of the mice did we observe the emergence of window occlusions that caused an obvious reduction of fUS imaging quality. We could not evaluate the lifetime of windows beyond the limit imposed by our ethics protocols (maximum 63 days); however, many of the experiments performed in the current work and described in later sections took place as late as 10 weeks after implant installation with consistent imaging quality for acoustic and optical imaging.

To verify the long-term biocompatibility of the COMBO window, we immunohistochemically examined brain slices 6 weeks after installation by visualizing reactive gliosis through glial fibrillary acidic protein (GFAP), a widely used marker for astrocytes and the immune response of the brain [28]. Overall, we found that chronic exposure to the COMBO window did not elicit a widespread inflammatory response in the cortex (**Fig 1F**). However, we anecdotally observed small and sparsely distributed patches of increased GFAP signal in implanted animals (**S4A and S4B Fig**) that were likely caused by localized surgical damage rather than chronic inflammation. In fact, an unbiased quantification of GFAP fluorescence in 18 superficial and uniformly distributed cortical regions-of-interest (ROIs) in each animal revealed no significant difference between those implanted with the COMBO window ($N$ = 6 mice, 6 weeks post-installation) and those without ($N$ = 5 mice) (**Fig 1F**). These results demonstrate that our design provides access to a large fraction of the mouse brain without causing long-lasting inflammation. We further investigated more global health effects of the COMBO window by tracking the weight of mice after installation. As expected with an invasive surgical procedure, compared to age-matched littermates ($N$ = 4 mice), those mice implanted with the COMBO window ($N$ = 6 mice) exhibited a significant reduction in weight within the first 3 days after installation (<10% loss) (**S5A**–**S5C Fig**). Nevertheless, the weight of implanted animals returned to presurgical and control levels within 4 to 5 days, and persisted even after attachment of the head plate, suggesting no long-term systemic effects.

## The COMBO window preserves freely moving behavior and facial expressions

We next characterized the impact of the COMBO window on various aspects of mouse behavior. Behavioral effects due to the added weight of the implant may not be apparent in head-fixed experiments, but could significantly affect an animal's locomotion in freely moving contexts (e.g., the home cage), and negatively affect their quality of life and induce stress over time [29]. Therefore, we assessed the effects of head plate weight on natural behavior during an open-field free foraging task (**Fig 2A**) [30]. For these experiments, we implanted wild-type C57BL/6 mice ($N = 7$ mice) with the "cup" version of the COMBO window and attached a 1.5-mm thick stainless-steel head plate (2.5 g) as an upper bound of added weight. Two weeks after head plate attachment, the mice were left to explore a 40 cm × 40 cm arena containing randomly placed food pellets for 10 min while their movement was tracked. We compared different locomotion-related variables between implanted ($N = 7$ mice) and control mice ($N = 7$ mice): all statistical tests and results are provided in **S1 Table**. We did not find any observable difference in open field coverage or cumulative distance traveled between the 2 groups of animals (**Fig 2B and 2C**). More specifically, there was no significant difference in the total distance traveled or locomotor speed (**Fig 2D and 2E and S1 Table**). We further assessed whether the unilateral head plate design and unbalanced weight distribution impaired the animals' ability to turn by measuring the tortuosity of their open field trajectory. The tortuosity was not significantly different between control and implanted mice (**Fig 2F and S1 Table**). Considering that female mice typically weigh less than male counterparts, we further inspected sex-dependent differences across the same parameters. We found no significant effect of sex for any of these parameters, and no difference between implanted and control mice within a sex (**S6A–S6C Fig and S1 Table**), indicating that these open field observations generalize to both male and female mice.

In addition to naturalistic behavior, we also sought to determine the effects of the implant on orofacial movements as the implant reaches the lateral-most edges of the skull and could impact the nearby skin and muscles. In recent years, head-fixed neural recordings have been commonly accompanied by the analysis of concurrent orofacial videography to link brain activity with the detailed assessment of behavioral states [24,31,32]. To verify that orofacial movements are preserved with the COMBO window, we tested the ability to identify facial readouts of emotion states which rely on stereotyped position patterns of the ears, snout, and whiskers [25]. We implanted wild-type C57BL/6 mice ($N = 3$ mice) with the COMBO window and subsequently habituated them to head-fixation on a running wheel. Mice were then provided with individual trials of sucrose and quinine in subsequent runs to elicit the emotions of pleasure and disgust, respectively (**Fig 2G**). These 2 stimulus-emotion pairings were chosen since the associated facial expressions comprise movements that span the full range-of-motion of the ears and snout. When examining the videographic frames identified as prototypical emotion states (see Methods for details), we observed in all 3 mice the stereotypical features of the 2 facial expressions (**Figs 2H and S6D and S1–S2 Videos**). As originally reported in mice with no cranial window [25], we found that the readout of pleasure and disgust, quantified as the temporal correlation with a prototypical facial expression of emotion, were highly selective for sucrose and quinine, respectively (**Fig 2I and 2J**). The disgust response during quinine trials was significantly larger than that during baseline, whereas it was nonexistent during sucrose trials (**S2 Table**). Accordingly, the pleasure response during sucrose trials was significantly larger than that during baseline trials, but was absent in quinine trials (**S2 Table**), demonstrating successful emotion identification from the orofacial movements of implanted animals.

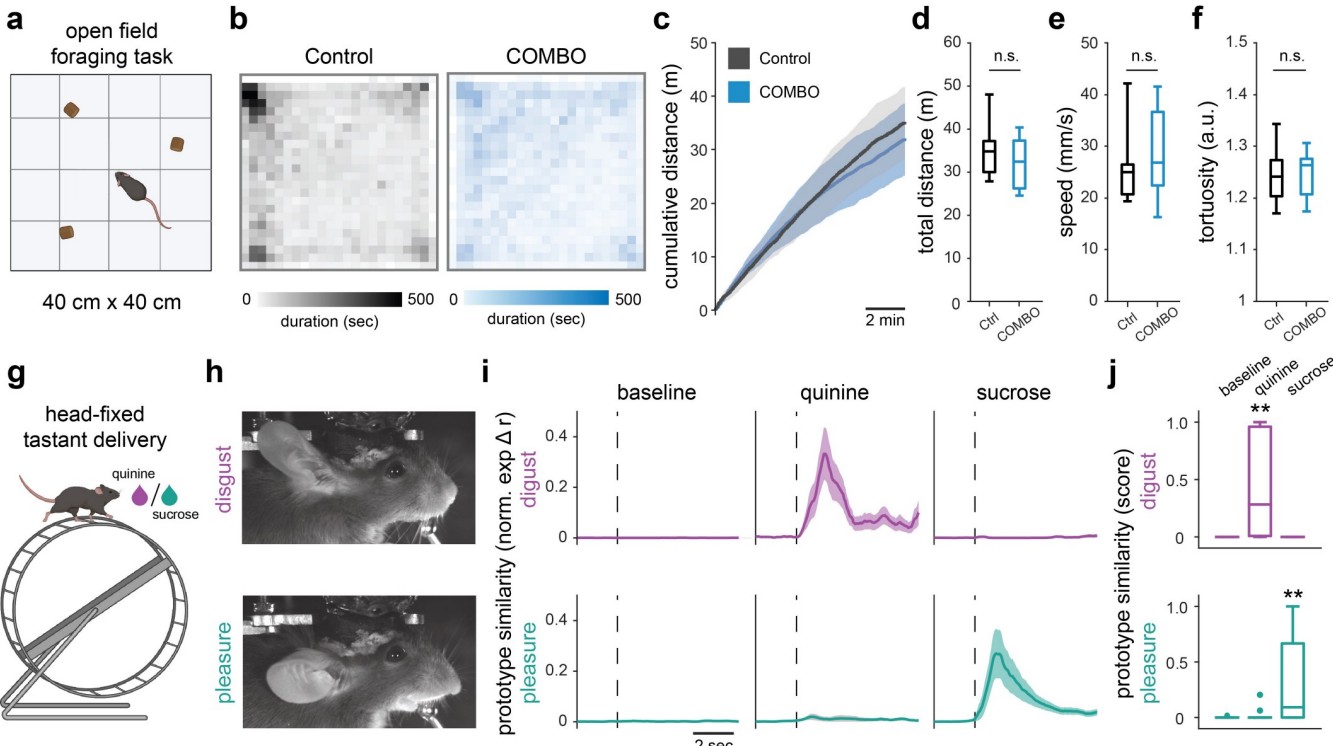

**Fig 2. The COMBO window preserves freely-moving behavior and facial expressions.** (**a**) Animals with a COMBO window (COMBO) (*N* = 7 mice) and control (ctrl) (*N* = 7 mice) performed an open field foraging task to test the effect of the implant and head plate on freely moving behavior. (**b**) Average trajectories of control mice (left, gray) and mice with a "cup" implant and 1.5 mm stainless steel head plate installed (right, blue) running in a 40 × 40 cm open field for 10 min. (**c**) Cumulative distance of the control and COMBO mice over the course of 10 min runs. (**d–f**) The total distance (**d**), median speed during running (**e**), and tortuosity during exploration (**f**) of control and COMBO mice. In all plots, black color represents control mice and blue color represents COMBO mice. Two-way ANOVA on ranks with main effects of sex and COMBO window. Main effect of the COMBO window (*N* = 7 mice per group): n.s. $p > 0.05$. (**g**) Head-fixed mice (*N* = 3 mice) were administered 5 trials of sucrose and quinine (2 s each) in separate runs to extract and evaluate the specificity of the pleasure and disgust facial readouts of emotions. (**h**) Individual videographic frames displaying a prototypical disgust and pleasure facial expression from a representative mouse with the "cup" version of the COMBO window. (**i**) Time-resolved normalized pleasure and disgust facial prototype similarity scores during baseline, sucrose, and quinine events (*n* = 15 trials each). Dark lines and light shaded areas represent the mean ± SEM across trials. (**j**) Trial-based normalized prototype similarity scores during baseline, sucrose, and quinine trials (*n* = 15 trials). Wilcoxon rank sum test between tastant and baseline events: ** $p < 0.01$. Boxplots represent the median (center line), 25th and 75th percentiles (lower and upper box), and the first and 99th percentile (whiskers). Additional dots represent outliers that fall below the first or above the 99th percentile. Images in **a** and **g** were created with BioRender.com. Underlying data can be found in **S2 Data** and code in **S2 and S11 Codes**. COMBO, ChrOnic Multimodal imaging and Behavioral Observation.

Overall, these results show that the COMBO window does not affect locomotor or orofacial behavioral readouts in freely moving or head-fixed contexts, respectively.

## Brain-wide fUS imaging through the COMBO window

We next aimed to demonstrate the utility of the COMBO window for volumetric brain-wide fUS imaging [23]. fUS imaging measures blood volume as a proxy of brain activity [33] and, in the context of multimodal imaging, gives access to large-scale information while being compatible with standard behavior rigs. We chose to use visual stimulation because the mouse visual pathway has been previously described in detail using both fUS and other large-scale imaging techniques [8,34]. We implanted wild-type C57BL/6 mice (*N* = 4 mice, *n* = 46 sessions) with the "cup" version of the COMBO window before habituating them to a head-fixed context in which the mouse is resting in a tube in front of a display screen (**Fig 3A**). At 4 to 6 weeks after installation, we acquired brain-wide fUS data while simultaneously presenting awake mice with drifting gratings moving along the 4 cardinal directions in the binocular

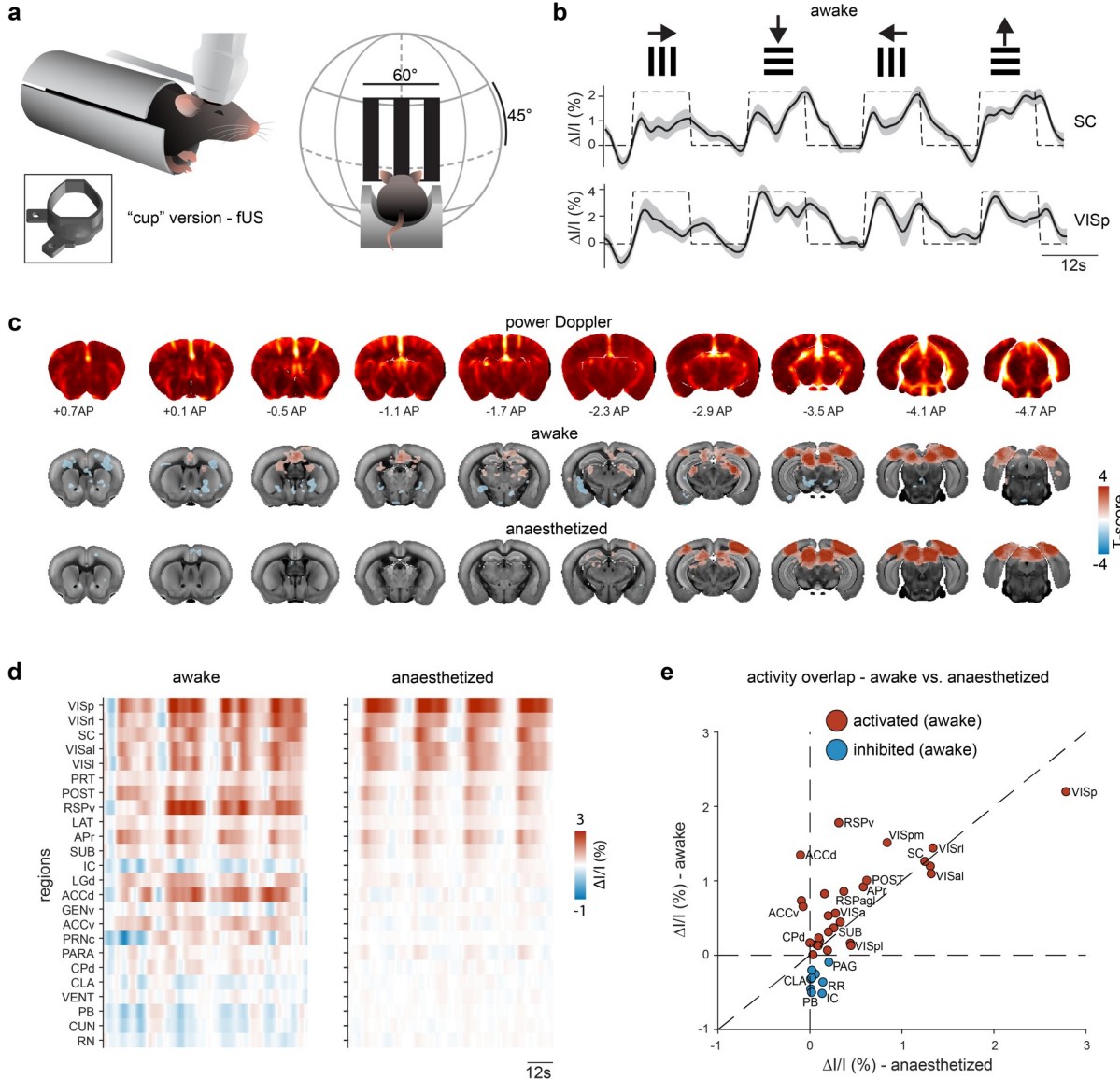

**Fig 3. Awake brain-wide imaging through the COMBO window.** (**a**) Schematic representation of fUS imaging in mice in a holding tube with the "cup" version of the COMBO window. Awake mice were presented with drifting gratings in the visual field (size 60° visual angle) while brain-wide fUS data was collected. (**b**) Evoked fUS signals were observed in the primary visual cortex (VIS) and SC in response to drifting gratings moving in all 4 cardinal directions. The dashed black line indicates the stimulus timing. The dark black line and light gray shaded area represent the mean ± SEM across sessions ($N$ = 4 mice, $n$ = 46 sessions). (**c**) Coronal slices of power Doppler images (top row) and T-maps (middle row: awake, bottom row: anesthetized) are overlaid on the Allen Brain Atlas at the indicated positions (anterior-posterior distance from bregma). Voxels with a significant T-score across sessions ($p < 0.05$, FDR-corrected) are displayed in color (awake: $N$ = 4 mice, $n$ = 46 sessions, anesthetized: $N$ = 5 mice, $n$ = 32 sessions). (**d**) fUS data was segmented into anatomical brain regions. Only regions that were significantly modulated by the visual stimulus in the awake state are displayed (correlation coefficient significantly different from zero across sessions, $p < 0.05$, FDR-corrected). (**e**) The mean fUS signal (ΔI/I) in each region (same as **d**) during stimulus presentation plotted as the anesthetized (x-axis) vs. awake (y-axis) state. Not every region label is shown for visualization purposes. Regions are colored according to the fUS signal in the awake state (blue circles for negative ΔI/I, red circles for positive ΔI/I). Underlying data can be found in **S3 Data** and code in **S3** and **S11 Codes**. COMBO, ChrOnic Multimodal imaging and Behavioral Observation; fUS, functional ultrasound; SC, superior colliculus.

visual field (**Fig 3A**). This visual stimulus specifically triggers oculomotor movements in awake mice [8]. Each stimulus block (12 s on, 12 s off) was presented 3 times in a randomized order for a total recording time of less than 5 min per session. Using an ultrasound transducer

specific to volumetric acquisition (called a "matrix probe"), we acquired volumetric fUS brain data at approximately 2 Hz temporal resolution [23]. We examined the hemodynamic response in core visual areas such as the primary visual cortex (VIS), lateral geniculate nucleus (LGN), and superior colliculus (SC). As expected, the fUS signal within these regions correlated with the visual block stimulus ($r_{VIS}$ = 0.68, $r_{SC}$ = 0.71) (**Fig 3B**) and the activity was robust at the level of individual trials and sessions, across animals (**S7A Fig**). We then assessed the brain-wide, voxel-wise responses using a general linear model (GLM) analysis. We observed significant bilateral activity in hubs of the visual system, including the VIS, SC, and LGN (**Fig 3C and 3D** and **S3 Video**). The symmetry indicates that the unilateral design of the COMBO window does not obstruct the perception of bilateral visual inputs. Additionally, as previously reported, we observed a decrease in amygdala activity (**Fig 3C and 3D**) due to the oculomotor movements elicited by the drifting gratings [8]. Lastly, a substantial increase in activity was found in the dorsal-medial striatum (DMS), the visual domain of the striatum [35], as well as the anterior cingulate cortex (ACC) (**Fig 3C and 3D**). The activation in these anterior regions further refines the characterization of brain-wide visual responses to these stimuli, which had not been observed in previous work due to a smaller and more posterior field of view [8]. The ability to detect visual-evoked activity in brain regions spanning more than 5,4 mm in the anterior-posterior direction and 8,4 mm in the medial-lateral direction highlights the utility of an implant that can accommodate such a large cranial window.

Imaging experiments in awake animals come with the advantage of enabling brain activity measurements during a large repertoire of natural behaviors. By contrast, imaging under anesthesia can improve data quality due to reduced animal movement and fewer motion-induced artifacts. Moreover, the investigation of sensory brain networks in anesthetized recordings is not confounded by neuronal activity elicited by ongoing behaviors. To compare the 2 approaches, we repeated the experiment in anesthetized mice ($N$ = 5 mice, $n$ = 32 sessions) using an identical visual stimulation paradigm. In this case, we expected a robust activation of the main visual pathway but no activation of the oculomotor-related pathways. Indeed, in anesthetized animals, we also observed robust evoked activity in the core visual regions (**Fig 3C and 3D**), such as VIS, SC, and LGN, with a similar evoked amplitude as in awake animals (**Figs 3B, 3D, 3E, S7B and S7C**). We also observed that a cluster of regions, including the ACC and subregions of the amygdalar complex, were modulated by visual stimulation in awake, but not anesthetized animals (**Fig 3C–3E**). Lastly, to demonstrate the durability of the COMBO window, we also collected brain-wide fUS data at 2 months postimplantation ($N$ = 1 mouse, $n$ = 8 sessions) and still detected robust activation in the core regions of the visual system (**S7D Fig**). Due to animal license limitations, mice were sacrificed after this time point; however, the windows remained acoustically viable. Together, these results demonstrate that the COMBO window enables longitudinal and stable brain-wide fUS imaging in behaving mice.

## The COMBO window enables multisite two-photon imaging during behavior

To assess the ability to capture single-cell activity through the COMBO window using optical imaging techniques, we created the "flat" version by removing the upward protruding cup (the film still shaped to the curvature of the brain), which allows closer positioning of the objective towards the brain surface (working distance of objective: 2 mm). In cases where longitudinal and multimodal imaging is preferred, the "flat" COMBO window can also be used with fUS imaging, as the "cup" simply helps to contain the ultrasound gel. The COMBO window was installed on mice ($N$ = 3 mice) expressing the genetically encoded $Ca^{2+}$ indicator GCaMP6s [36] in cortical excitatory cells. At 10 weeks postimplantation, we leveraged the large cranial

window by imaging Ca$^{2+}$ signals at a depth of 200 to 300 μm (layer 2/3) in the secondary motor (M2) and retrosplenial (RSC) cortices, 2 brain areas located at the anterior and posterior edges of the cranial window. Across these mice, we recorded Ca$^{2+}$ signals from a total of 150 neurons in M2 and 180 neurons in RSC. During two-photon imaging, the animals were head-fixed on a running wheel and spontaneous movements were captured using videographic recordings (**Fig 4A**). To directly assess the stability and suitability of the COMBO window for two-photon recordings during behavior, we examined pre- and post-registration metrics via the *Suite2p* processing framework [37]. As seen in similar experimental setups [38,39], we observed larger frame-wise displacements in both the x and y directions during periods of increased locomotion (**Fig 4B**). To measure the ability to successfully recover cellular traces, we examined the residual drift of the spatial principal components extracted by *Suite2p* after non-rigid registration. The residual drift value for both the M2 (median = 0.27 μm, IQR [0.23, 0.49]) and RSC (median = 0.10 μm, IQR [0.03, 0.12]) recordings was well below 1 μm (**Fig 4C**), and is similar to the values reported by recently benchmarked two-photon imaging studies [30,37]. This shows that the COMBO window provides sufficient stability for high-resolution two-photon imaging across distributed cortical areas in behaving animals.

While the previous analysis demonstrates stability from a data analytical standpoint, we further ensured that the COMBO window enables the acquisition of neural activity associated with head-fixed mouse behavior. For this, we took advantage of a simple but robust phenomenon whereby a significant portion of neural activity in the mouse cortex can be explained by behavioral variables such as locomotion [24,40]. Specifically, we performed keypoint tracking of the forepaws from videographic recordings (see Methods for details) and compared a time-resolved output of locomotion to neuronal traces in both M2 and RSC (**Fig 4D and 4E** and **S4 Video**). As demonstrated in various large-scale two-photon studies, we found that the first principal component (PC) of population activity was highly correlated with locomotion in both M2 (median $r$ = 0.70, IQR [0.27, 0.79]) and RSC (median $r$ = 0.55, IQR [0.28, 0.71]). Similarly, at the individual neuron level, we found that the majority of cells were significantly correlated with locomotion in both recording sites (84% in M2, 85% in RSC, **Fig 4E and 4F**). More specifically, we observed a positive correlation with locomotion for the majority of M2 cells (positive: 61%, negative: 23%), while positively and negatively correlated cells represented a similar proportion of all neurons in RSC (positive: 41%, negative: 44%) (**Fig 4F**). Nevertheless, in both regions a small fraction of cells was neither positively or negatively linked to locomotion (16% in M2, 15% in RSC). Together, these results demonstrate the utility of the COMBO window for optical imaging at cellular resolution across the mouse cortex during head-fixed behavior.

## Multimodal investigation of optogenetic circuit perturbations

Optogenetic approaches are powerful tools to manipulate genetically identified cell types or brain regions. We aimed to create a framework that enables the identification of brain-wide activity patterns in response to specific optogenetic manipulations in behaving animals, as well as the subsequent verification of these patterns using alternative invasive techniques such as electrophysiology in the same animals. Longitudinal studies that enable within-animal verification are particularly useful for optogenetics, which can exhibit high variability in opsin expression. To increase the flexibility of this framework, we designed 7 additional versions of the COMBO window to provide different "off-the-shelf" options for different experimental needs (**Fig 1D**). For validation of this design principle in awake animals, we targeted the secondary motor cortex (M2), as optogenetic activation of this region has been previously shown to cause a robust increase in locomotion [41]. We bilaterally injected an adeno-associated virus carrying the channelrhodopsin-2 (ChR2) construct under the

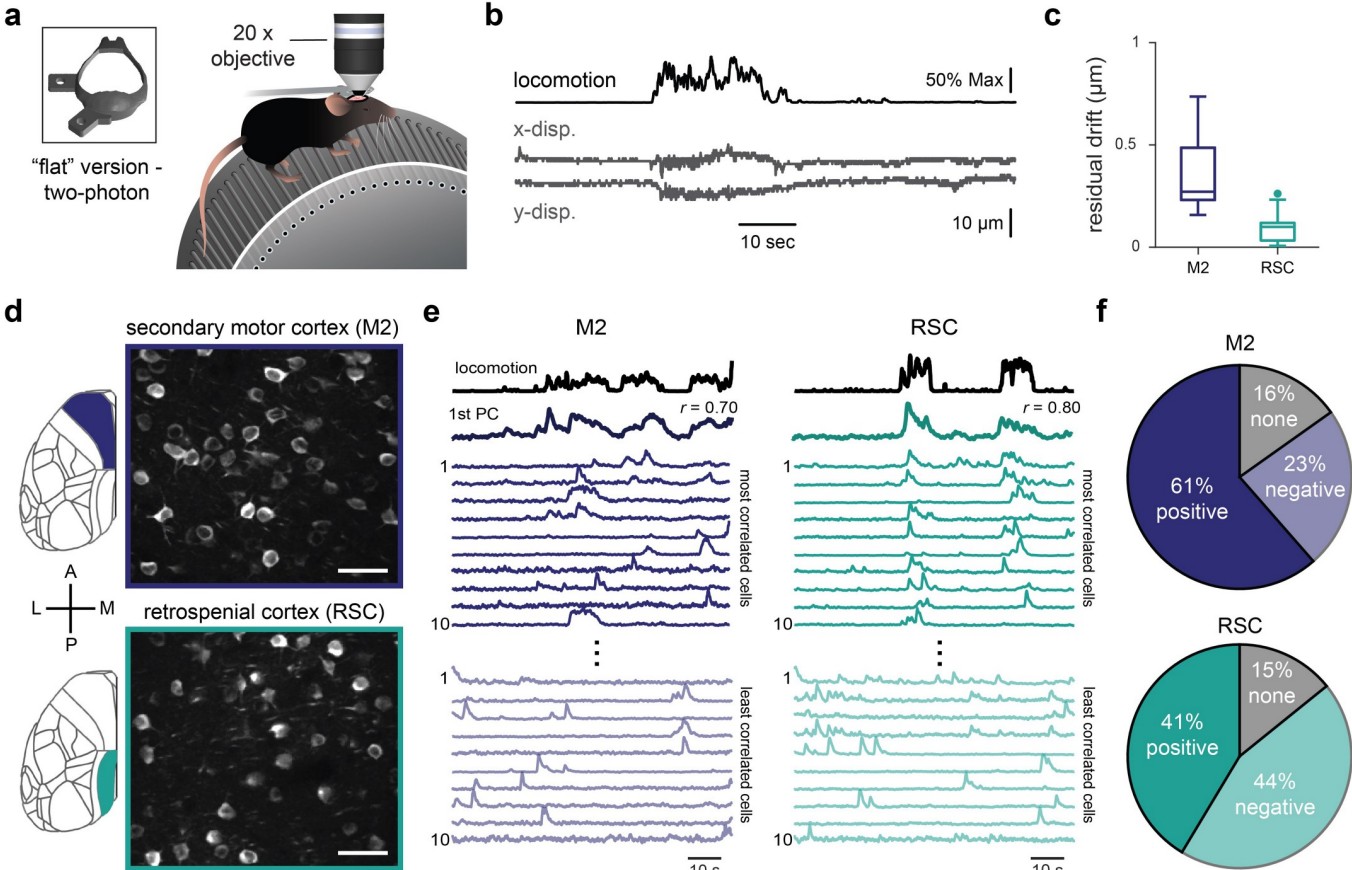

**Fig 4. The COMBO window enables multisite two-photon imaging during behavior.** (**a**) Schematic of two-photon Ca²⁺ imaging in mice on a running wheel with a "flat" version of the COMBO window. (**b**) Example locomotion trace along with the framewise displacement of images in the x and y direction during a representative 90-s recordings session. (**c**) Residual drift of secondary motor cortex (M2) (*N* = 3 mice, *n* = 8 sessions) and RSC (*N* = 3 mice, *n* = 7 sessions) recordings after non-rigid registration of the images. Boxplots represent the median (center line), 25th and 75th percentiles (lower and upper box), and the first and 99th percentile (whiskers). Additional dots represent outliers that fall below the first or above the 99th percentile. (**d**) Representative maximum intensity projection of a recording acquired 200–300 µm below the pial surface in M2 (top, purple) and RSC (bottom, turquoise). Scale bars represent 50 µm. (**e**) Example recording from M2 (left) and RSC (right), showing animal locomotion (first trace) and the first PC of the neuronal population activity (second trace) as well as the times series from the 10 cells most (middle traces) and least (lower traces) correlated with locomotion. (**f**) Pie charts representing the proportion of individual cells significantly correlated with locomotion in M2 (150 cells, *N* = 3 mice, *n* = 8 sessions) and RSC (180 cells, *N* = 3 mice, *n* = 7 sessions). *P*-values for each cell-locomotion correlation combination were FDR-corrected and thresholded at *p* < 0.001. Remaining cells were then categorized as positive (dark color) or negative (light color) based on the sign of the correlation. Cells with a *p*-value > 0.001 were categorized as none (gray). Underlying data can be found in **S4 Data** and code in **S4 Code**. COMBO, ChrOnic Multimodal imaging and Behavioral Observation; PC, principal component; RSC, retrosplenial cortex.

*Camk2a* promoter (AAV9-CaMKIIa-hChR2(E123A)-EYFP) into M2 in wild-type C57BL/6 mice (*N* = 5 mice). In addition, optic fibers were implanted and a cranial window was created, all of which were encapsulated and stabilized using the "anterior" version of the COMBO window (**Fig 5A and 5B**). Transduction was verified at the end of the experiment using immunohistochemistry, which confirmed strong transgene expression in the bilateral M2 (**Fig 5B**). We first analyzed behavioral responses elicited by optogenetic activation of M2 using motion energy analysis of simultaneously acquired videographic recordings (**S8A Fig**). As expected, 10 s of blue light at 20 Hz elicited a robust increase in locomotion (peak z-score: 8.25 ± 1.89) (**Fig 5C and 5D**). This stimulation was also associated with an increase in whisking (peak z-score: 1.86 ± 0.40), a more detailed behavior that could be captured due to the unilateral design of the COMBO window (**Figs 5D and S8B**). To control for potential light-induced effects, we also injected a non-ChR2 carrying virus (AAV9-CaMKIIa-EGFP)

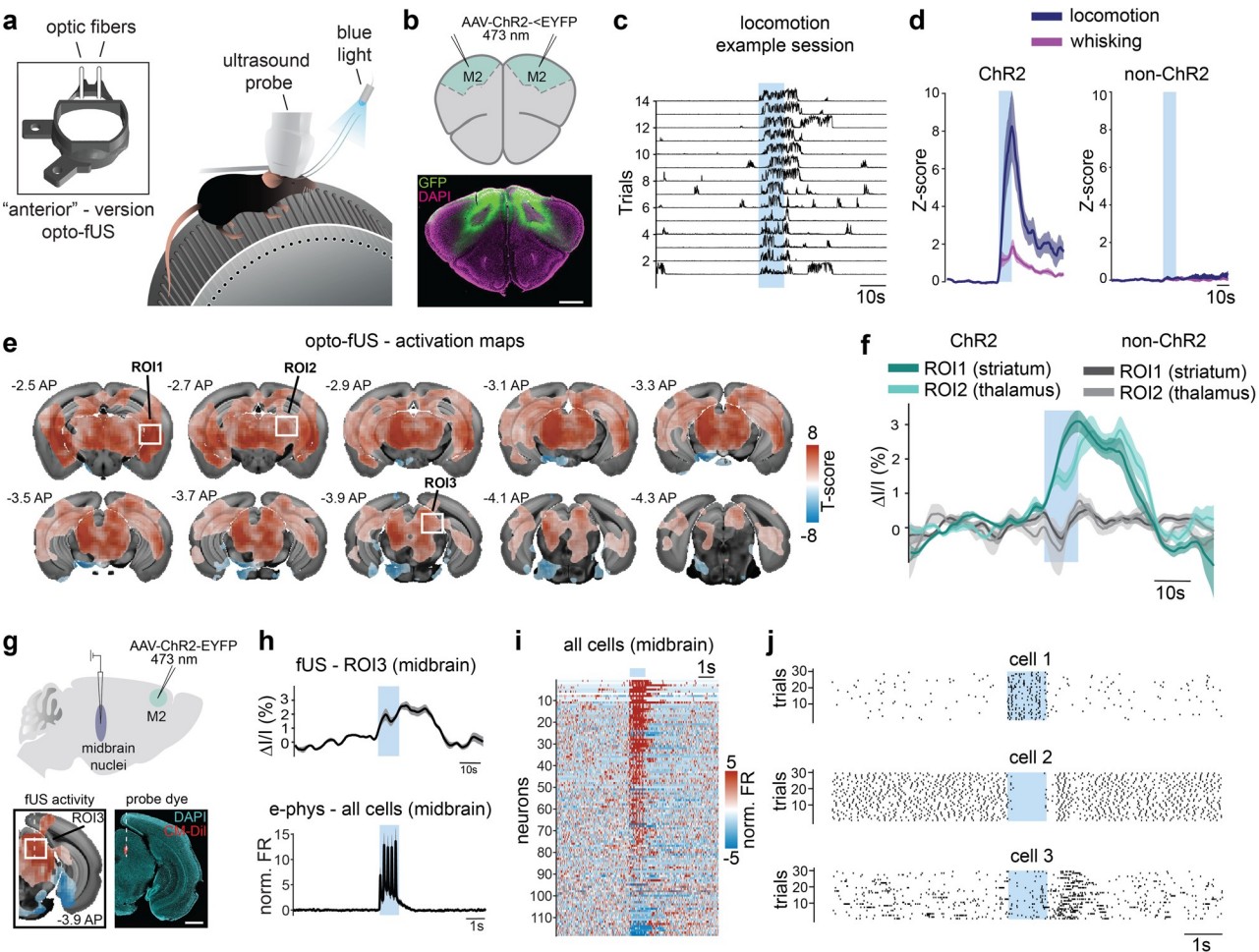

**Fig 5. Multimodal investigation of optogenetic circuit perturbations.** (**a**) Schematic representation of simultaneous fUS imaging and optogenetic stimulation (473 nm) of secondary motor cortex (M2) in mice with the "anterior" version of the COMBO window on a running wheel. (**b**) Before window installation, AAV9-hChR2(E123A)-EYFP was injected bilaterally into M2, resulting in robust expression of hChR2(E123A)-EYFP (EYFP in green and DAPI in magenta). Scale bar represents 1 mm. (**c, d**) Behavioral parameters were assessed by quantifying motion energy changes from wheel and whisker ROIs. (**c**) Consecutive trials from 1 session after applying a z-score transformation to the pre-stimulus baseline. Traces are rescaled between 0 and 1 for visualization purposes. (**d**) Optogenetic activation of M2 results in a robust increase in locomotion and whisking (locomotion: 8.25 ± 189 peak z-score; whisking: 1.86 ± 0.40 peak z-scores (mean ± SEM) ($N$ = 5 mice, $n$ = 22 sessions) (left). These effects were absent in non-ChR2 expressing control animals (locomotion: 0.3 ± 0.18 peak z-score; whisking: 0.14 ± 0.13 peak z-score) ($N$ = 3 mice, $n$ = 12 sessions) (right). (**e**) Average T-maps are overlaid on the Allen Brain Atlas at the indicated positions (anterior-posterior distance from bregma). Voxels with a T-score significantly different from zero across sessions ($p$ < 0.05, FDR corrected) are displayed in color ($N$ = 3 mice, $n$ = 16 sessions). White square boxes represent ROIs used to extract fUS time series data in **f** and **h**. (**f**) Group-level fUS traces from the striatum and thalamus for ChR2-expressing animals (green, $N$ = 3 mice, $n$ = 16 sessions) and non-ChR2 expressing control animals (gray, $N$ = 3 mice, $n$ = 12 sessions). (**g**) Electrophysiological recordings were performed during optogenetic stimulation of M2 to verify the fUS signal. Silicon probes successfully targeted the midbrain (CM-Dil dye in red and DAPI in cyan), where strong optogenetically induced fUS activation was observed. Scale bar represents 1 mm. (**h**) The mean normalized firing rate across all recorded cells in the midbrain (bottom) compared to the average fUS signal (top) of ROI 3 (midbrain) from **e**. (**i**) The normalized firing rate of all recorded cells sorted by mean activity during optogenetic stimulation. (**j**) Raster plots showing all spikes from a given session for 3 example cells, highlighting the diversity of single-cell responses. For all panels, blue squares indicate the time of optogenetic stimulation. Solid lines and shaded areas represent the mean ± SEM across sessions or cells. Underlying data can be found in **S5 Data** and code in **S5 and S11 Codes**. COMBO, ChrOnic Multimodal imaging and Behavioral Observation; fUS, functional ultrasound; ROI, region-of-interest.

into M2 of wild-type C57BL/6 mice ($N$ = 3 mice) and applied the same stimulation protocol. In these control mice, we did not observe a noticeable effect for locomotion (peak z-score: 0.3 ± 0.18) or whisking (peak z-score: 0.14 ± 0.13) (**Fig 5D**).

To observe the brain-wide activity associated with optogenetic activation of M2, we also simultaneously acquired volumetric fUS recordings in a subset of ChR2-injected mice (*N* = 3 mice, *n* = 16 sessions). We found robust stimulation-locked activated voxels throughout the brain (**Fig 5E**). Specifically, we observed strong activation in the thalamus, the midbrain, and posterior/medial subregions of the striatum (**Figs 5E, 5F, 5H and S8C**). Qualitatively, we observed a good correspondence of the observed fUS activity with documented M2 axonal projection patterns described in literature [42] as well as in the Allen Brain Connectivity Atlas (**S8D Fig**) [43]. To better understand the extent by which the observed brain activation patterns are driven by optogenetically elicited behaviors, we exploited the natural variability of the recorded behavioral responses. More precisely, we identified trials that elicited strong bouts of locomotion and others that did not (see Methods for threshold definition). By doing so, we compared the fUS activity during optogenetic stimulation in the presence and absence of locomotion. Overall, we found that the fUS activation patterns were qualitatively similar between the 2 conditions (**S8E and S8F Fig**), suggesting that a large portion of the observed brain activity was a direct result of M2 stimulation and not an indirect consequence of the elicited behavior. Additionally, as light-induced vasodilation has been previously reported [22,44,45], we examined in detail the responses in awake "non ChR2" mice (*N* = 3 mice, *n* = 12 sessions) and recorded additional sessions under anesthesia (*N* = 3 mice, *n* = 14 sessions) to maximize the detectability of potential light effects. Analogous to the behavioral analysis, we did not find any significant evoked neural activation in response to stimulation in control mice ("non-ChR2") for either awake or anesthetized contexts (**Figs 5F and S9**), demonstrating that the observed neural responses are due to optogenetic M2 activation. These results together demonstrate that the COMBO window enables the observation of a large volume of the brain during bilateral optogenetic stimulation in behaving animals.

One additional advantage of using a polymer-based film to seal the implant, especially in contrast to glass coverslips, is the potential to directly access the brain using more invasive techniques. To demonstrate this feature, we acquired electrophysiological recordings in a subset of injected and implanted mice (*N* = 2 mice) in response to optogenetic activation of M2. For this, we drilled small perforations (0.5 mm diameter) in the film and lowered 32-channel silicon probes into the midbrain at coordinates where we previously observed strong fUS activation (AP: −4.00 mm, DV: −2.20 to −3.60 mm, ML: ±1.0 mm) (**Fig 5E, 5G and 5H**) and optogenetically activated M2 at 5 Hz. We recorded 118 single units across the midbrain reticular nucleus and the motor-related superior colliculus (**Fig 5H and 5I**). In correspondence with our fUS results and the known M2 axonal projection patterns to the midbrain [42,43], we observed a strong increase in firing rate at the population level during M2 activation (peak z-score: 17.87 ± 3.58 z-scores, **Fig 5H**). In total, stimulation-induced modulation of the firing rate was observed in 86% of the recorded cells, verifying the local fUS signal (**Figs 5I and S8G**). More specifically, 72% of cells responded with a short-latency increase in firing rate (**Figs 5J cell 1 and S8G**), 9% with a decrease in firing rate (**Figs 5J, cell 2 and S8G**) and 6% with an increase in firing rate at the offset of stimulation (**Fig 5J, cell 3 and S8G**). Overall, this proof-of-concept experiment shows the compatibility of the COMBO window with electrophysiological recordings and highlights the overall versatility of the design.

## Discussion

In this work, we present a whole-brain cranial window implant for the independent and combined application of fUS imaging, optical imaging, optogenetics, electrophysiology, and behavioral observations in head-fixed mice. We show that the COMBO window is well tolerated by implanted animals and preserves both detailed orofacial movements (e.g., emotion readouts)

and natural freely moving behavior. In separate cohorts, we demonstrated the feasibility of using fUS and optical imaging to obtain brain-wide (fUS) and single-cell (two-photon) measurements of neural activity, respectively, through the COMBO window in behaving mice. Finally, we demonstrated that the COMBO window can be used to observe the brain-wide effects of optogenetic circuit manipulations in head-fixed mice. Importantly, the COMBO window allowed us to perform an array of different recordings and manipulations all in the same animals. For example, here we utilized the COMBO window to simultaneously observe whole-brain activity patterns and elicited behaviors induced by targeted cell type-specific activation of M2, as well as the subsequent electrophysiological recordings in targeted brain regions. An open-source design and detailed protocols are provided to facilitate the adoption of the COMBO window by neuroscience labs (**S1**–**16 Files, S1** and **S3 Figs, S1 Appendix, https://doi.org/10.5281/zenodo.11092491**).

To achieve compatibility of the implant with multiple techniques, we chose a polymer-based design for the COMBO window. Compared to glass, plastic does not attenuate ultrasound and allows small perforations to be made for electrophysiology. Additionally, the flexibility of plastic enables a tight fit to the curvature of the exposed mouse brain and promotes a low impact on the brain and long-lasting window transparency. The COMBO window provides a ~90 mm$^2$ opening, which is a notable increase in field of view over alternative implant designs that provide access to 45–75 mm$^2$ of the dorsal cortex [16,18], as well as an improvement in reproducibility compared to other preparations that require window customization for each animal [20,46,47]. With this being said, the larger cranial window associated with the COMBO window comes with the cost of potentially reduced stability (see below) and a more difficult surgical procedure. Despite the large size of the cranial window and in contrast to many other preparations [8,20,27,46,47], we ensured that the COMBO window and custom-made head-fixation allow unobstructed readouts of mouse behavior, which constitutes a growing need in system neuroscience studies. Fine analysis of facial movements, for example, are widely used to characterize arousal or emotion states in head-fixed animals [25,48]. Additionally, compared to other previously published implant designs, we provide off-the-shelf solutions for many combinations of neural interrogation tools that make custom adjustments by the users, i.e., due to different regions of interest, unnecessary.

The proof-of-concept experiments performed in this study were specifically designed to investigate the usability of the COMBO window; however, they also suggest new avenues for addressing future neuroscientific questions. First, we demonstrate that the COMBO window enables brain-wide imaging during behavior without any noticeable effects on mouse locomotion or orofacial movements. This unique advantage is further strengthened by the stability provided by the implant. For example, we observed consistent activation of the main visual pathways in both awake and anesthetized states. We also identified visual-evoked activity in the striatum and ACC in awake but not anesthetized mice, potentially reflecting a state-dependent recruitment of brain regions. This is in contrast to a similar comparison study performed using fMRI where a loss of apparent cortical activation was reported in the awake state compared to the anesthetized state [34]. This discrepancy may be due to differences in the mode of visual stimulation (drifting gratings versus flashing LED) or in the technical sensitivities between the two methods [20,27]. Overall, the acquisition of state-dependent and brain-wide activation patterns in awake animals, as we demonstrate with the COMBO window, enables the investigation of neuronal substrates underlying behavior. When paired with the preserved readouts of naturalistic behaviors, this framework suggests that the COMBO window can be used to study brain states that extend beyond the sensory domain.

An important feature of the COMBO window is its utility for measuring brain activity at different spatiotemporal scales with different modalities. Importantly, this carries the

advantage of enabling the conversion of many cross-sectional studies to longitudinal/paired designs, which can markedly help minimize the number of animals used and maximize statistical power [19,20,27]. Here, we specifically validated the COMBO window for compatibility with a number of commonly used neuroscience tools; however, it can also likely be paired with additional techniques. Foremost, as we confirmed the optical transparency of the window using two-photon imaging, we assume that other optical imaging techniques such as widefield calcium imaging are also compatible. Furthermore, similar to our proof-of-concept optogenetic stimulation experiment, other techniques requiring direct access to brain tissue, such as fiber photometry, microcannulas, or brain temperature sensing can also be implemented. However, it should be noted that the implantation of items required for these techniques reduces the potential size of an accompanying cranial window. Compatibility with (f)MRI is also theoretically possible, similar to what has been previously shown with other preparations [20,46] when imaging before the installation of the head plate. We provide a version of the COMBO window that is compatible with state-of-the-art MR receive coils for interested users. As such, the multimodal nature of the COMBO window could also be exploited in studies exploring whole-brain networks using resting-state functional connectivity [49]. In this context, the combination of multiple techniques with different neuronal readouts, such as calcium imaging, fMRI, fUS, and electrophysiology, potentially even in the same animals could help to extract complementary aspects of brain networks and their organizing principles across scales [50,51]. In fact, as we demonstrate in the current work, such brain networks may be further dissected by exploiting the COMBO window and its compatibility with optogenetic stimulations. In this context, however, researchers should exercise caution when interpreting optofUS results since local vasodilation has previously been observed with both fUS and fMRI in response to direct light stimulation at intensities as low as 2 mW [22,44,45]. In the current work, we ensured that no unspecific light effects contaminated our results by repeating the optogenetic stimulations in control animals that only expressed GFP in M2 (non-ChR2) in both awake (same as the ChR2-positive group) and anesthetized states. In the latter, behavior-induced variability and noise in the fUS signal is minimal; in these optimal conditions and with a relatively high light intensity (5 to 6 mW), we did not observe any significant light-induced activation. We could not image the stimulation site directly and therefore cannot rule out local effects; however, these results suggest that the observed brain-wide activity in response to optogenetic M2 activation in ChR2-expressing animals was not caused by light-induced vasodilation.

With large cranial openings, the concept of motion-induced artifacts becomes important as the surrounding skull usually stabilizes smaller windows. While our awake fUS data contained higher noise than anesthetized recordings, standard processing steps (e.g., scrubbing and removal of the first principal component, see Methods) revealed visual-evoked fUS activity in core visual brain regions. Similarly, optogenetic stimulation of M2 evoked expected patterns of activation that were confirmed at the neuronal level with electrophysiological recordings. These results confirm that motion artifacts can be efficiently removed while preserving the slow hemodynamic response elicited by sensory or optogenetic stimulations, even at the single trial level, albeit with more variability than during anesthesia. In our awake two-photon recordings, we also observed increased framewise displacement during periods of locomotion; however, image stacks were easily recovered using standard registration procedures implemented in open-source analysis toolboxes [37]. As with any head-fixed experiment, habituation to the fixation is an important step for animal comfort and for minimizing motion-related noise. As explained in our protocol (**S1 Appendix**), we find that at least 5 days of habituation, increasing the duration of head-fixation each day, is sufficient for stable fUS or two-photon recordings using the COMBO window. In the future, a version of the COMBO

window that allows for direct attachment of the imaging device (ultrasound probe, miniscope, etc.) to the head plate [52,53] could further reduce the effects of movement on data quality. Such a version could also be integrated in freely moving contexts and thereby expand the repertoire of behaviors that can be investigated. Nevertheless, the COMBO window provides a flexible, standardized, and open-source solution to combine various neural recording and manipulation techniques in awake head-fixed mice.

## Materials and methods

### Ethics statement

All experiments were carried out in compliance with institutional guidelines of the Max Planck Society and of the local government (Regierung von Oberbayern) under license numbers 55.2-2532.Vet_02-21-190, 55.2-2532.Vet_02-21-14, 55.2-2532.Vet_02-20-123.

### Animals

Male and female C57BL/6 mice (8 weeks old) were used for all fUS and behavior experiments. For two-photon calcium imaging, we used male mice (8 weeks old) that expressed GCaMP6s in excitatory cortical neurons (B6.DBA-Tg(tetO-GCaMP6s)2Niell/J [Jax, 024742] x B6.Cg-Tg (Camk2a-tTA)1Mmay/DboJ [Jax, 007004]). All animals were group-housed in a 12-h reversed light-dark cycle and were provided with standard diet and water ad libitum. All behavioral experiments were performed during the dark cycle.

### COMBO window preparation

COMBO window frames were printed using a stereolithography 3D printer (FormLabs, Form 2) using black resin (FormLabs). Minimal supports were placed with a 0.5 mm touchpoint size to ensure sufficient printing stability and to facilitate easy detachment afterward. Remaining support material was manually filed down to a smooth surface. The head plate attachment holes were threaded for M1.4 screws and 125-μm thick polymethylpentene (Goodfellow) film was attached to the interior ridge of the frame using a combination of cyanoacrylate glue (Pattex) and epoxy. The head plate was laser cut from 1.5-mm thick 304L stainless steel with a brushed finish to facilitate dental cement adhesion. The head plate was attached using both M1.4 screws and dental cement after the animals fully recovered from the cranial window surgery. Further details can be found in S1 Appendix.

### Cranial window surgery and COMBO window installation

Mice were anesthetized with a subcutaneous injection of a fentanyl (0.05 mg/kg), midazolam (5 mg/kg), and medetomidine (0.5 mg/kg) (FMM) cocktail. Mice were then secured using a bite bar and placed on top of a temperature controller (Supertech) to maintain a body temperature of 37˚C. Hydration gel (Bayer, Bepanthen) was placed on the eyes to prevent dryness during surgery. A dental drill was then used to cut a large cranial window into the skull, which generally spanned from bregma +2.25 mm AP to bregma −4.00 mm AP and the full width of the dorsal skull. The bone island was removed and the dura was left intact. The pre-prepared COMBO window was then attached to the remaining bone using cyanoacrylate glue (Pattex) and sealed with dental cement (Super-Bond). After surgery, anesthesia was reversed with a subcutaneous injection of a flumazenil (0.5 mg/kg) and atipamezole (2.5 mg/kg) cocktail. Carprofen (20 mg/kg) or Buprenorphine (0,1 mg/kg) was injected subcutaneously for postsurgical analgesia and was provided in case of pain during the recovery period. After 7 days of recovery, animals were again anesthetized with FMM as previously described and the head plate was

attached to the COMBO window. After at least 3 more days of recovery animals began habituation to the experimenter, the behavioral setups, and tasks.

## Virus injection and fiber implantation

For animals used in optogenetic experiments, virus injection and fiber implantation were performed prior to the craniotomy in the same surgery. Animals were anesthetized using the FMM cocktail previously described and secured in a stereotaxic frame (Stoelting). Mice were then injected bilaterally with 500 nl of AAV9-CaMKIIa-hChR2(E123A)-EYFP (Addgene, #35505, $2.2 \times 10^{13}$ particles per ml) or AAV9-CaMKIIa-EGFP (Addgene, #50469, $2.3 \times 10^{13}$ particles per ml) in the secondary motor cortex (AP: 2.50 mm, DV: 1.25 mm, ML: +/− 1.00 mm) at an angle of 30 degrees from the vertical. Custom-made optic fibers (200 μm core, 0.22 NA) were then implanted 200 μm above the injection site along the same trajectory. A minimal amount of cyanoacrylate glue was applied to the fibers and skull for stabilization while paying special attention to not cover the area intended for the cranial window. Then, the craniotomy was performed and the "anterior" COMBO window installed as previously described. In accordance with the smaller opening of the COMBO designs meant to accommodate optic fibers, these cranial windows generally spanned from bregma −1.50 mm AP to bregma −4.00 mm AP. The fiber compartment was then also filled with dental cement to ensure robust attachment to the skull.

## Optogenetic stimulation

All optogenetic stimulation experiments took place 2 to 3 weeks after virus injection to allow for sufficient expression. A 200-μm diameter optical fiber (Doric Lenses Inc.) was connected to a 473 nm laser (LaserGlow) and to each of the implanted fibers. The laser power was adjusted to 5 to 6 mW at the tip of each fiber. For experiments with simultaneous fUS and videography, a single stimulation run consisted of a block design with 90 s of baseline, followed by 15 trials of 10 s of optical stimulation (20 Hz, 10 ms pulse width) delivered every 90 s. Similarly, for optogenetic stimulation during electrophysiological recordings, a block design with 9 s of baseline, followed by 1 s of optical stimulation (5 Hz, 10 ms pulse width) was delivered every 9 s. Stimulation was triggered using a pre-programmed pulse generator (Doric, OTPG_8), which was synchronized with the fUS acquisition software, videographic recordings, and/or electrophysiological recordings, respectively.

## Visual stimulation

For visual stimulation, full-field drifting gratings were presented with a spatial frequency of 20˚ and velocity of 10˚/sec on a standard 61 cm computer monitor (Dell, U2415b) using the PsychoPy toolbox. The monitor was placed 18 cm in front of the mouse. We employed a block design consisting of 12 s of gray background followed by 12 s of drifting gratings (1 of 4 cardinal directions). Each direction was repeated 3 times, resulting in 12 stimulation blocks per recording run.

## fUS acquisition

fUS imaging data was acquired using a $32 \times 32$ channel matrix probe (15 MHz, 1,024 total elements, spatial resolution: $220 \times 280 \times 175 \mu m^3$, Vermon) attached to a Vantage 256 (Verasonics, Inc.) and controlled by a custom vfUSI acquisition module (AUTC) [23]. A 4× multiplexer was used to connect the 1,024 channel probe to the 256-channel system, with the beamforming and sequences adapted accordingly. At the beginning of each imaging session,

the matrix probe was positioned to encompass the full width and length of the cranial window using a three-way translation stage. Briefly, a single compound ultrasound image was generated from the summation of the reconstructed echoes of plane wave emissions at −4.5, −3, −1.5, 0, 1.5, 3, 4.5 degrees. A single power Doppler image was created from the incoherent average of 160 compound ultrasound images acquired at a pulse repetition frequency of 400 Hz. Clutter filtering was performed in real time whereby the compound ultrasound stack was decomposed using singular value decomposition and the first 20% of singular vectors were removed. This procedure produced a single power Doppler image every ~500 ms.

### fUS preprocessing

The fUS time series was first registered to the reference mouse brain atlas from the Allen Brain Institute. To do so, 100 fUS frames from a single recording session were first averaged together to create a higher resolution power Doppler image. This high-quality image was then manually registered (rotation and translation only) to the atlas using anatomical landmarks for a mouse-specific transformation matrix. This transformation matrix was then applied to data from other sessions from the same mouse. fUS time series were then preprocessed using custom MATLAB scripts on a voxel-by-voxel basis. First, data was temporally interpolated to obtain a constant frame rate of 2 Hz. Next, the relative change in power Doppler signal was calculated by first subtracting the baseline signal (mean of the 11 frames for data in Fig 3 or 70 frames for data in Fig 5 before each stimulus) from each time point and then dividing the result by the baseline signal. Subsequently, to remove slow drifts, the data was filtered with a fifth-order highpass Butterworth filter with a cutoff frequency of 0.056 Hz. Lastly, to remove unspecified movement artifacts during awake experiments, we removed the first principal component from the spatial-temporal decomposition of the entire time series. In addition, we implemented additional motion artifact rejection according to a previously validated approach [54]. For this, we removed motion artifact time points by interpolating those that were above a voxel-specific threshold, defined as the median + absolute deviation of signal * 4.44.

### fUS activation

For voxel-wise analysis, preprocessed fUS data was analyzed with a GLM analysis on a voxel-wise basis. First, a temporal smoothing of 4 frames was applied to the time series of each voxel, which was then fitted with a GLM using the MATLAB glmfit.m function. Model regressors included the optogenetic or visual stimulation block stimuli, after being convolved with a single-gamma hemodynamic response function. For each voxel, a one-sample $t$ test on the resulting T-scores from different sessions was performed and voxels with FDR-corrected $p$-value below 0.05 are displayed. The statistics were calculated across sessions since few animals were used for the presented proof-of-concept experiments.

For region-based analysis in Figs 3, S7 and S8 preprocessed and trial averaged whole-brain data was segmented into individual brain regions. Anatomical regions from the Allen reference brain atlas were condensed into 100 brain regions and used to segment the fUS time series by averaging the information from all voxels within a single region. To define active brain regions in response to visual and optogenetic stimulation, the correlation coefficient between the stimulus timing and the fUS signal of a brain region was calculated and a one-sample $t$ test on these correlation coefficients for each region was performed. All regions with an FDR-corrected $p$-value below 0.05 (0.01 for S8C Fig) were determined active. To visualize brain-wide activity across time in S3 Video, voxels and time points with low activity ($\Delta I/I < 1.75\%$) or high mean baseline activity ($\Delta I/I > 5\%$) were masked.

## Two-photon microscopy acquisition

$Ca^{2+}$ imaging was performed with a two-photon moveable objective microscope (Sutter Instruments) using a 20× objective (NA 1.0, Zeiss) and ScanImage software. GCaMP was excited at 940 nm using a Ti:sapphire laser (Mai Tai, Spectra Physics) and emission was detected using a GaAsP photomultiplier tube (Hamamatsu) through a bandpass emission filter (525/70 nm). To minimize photobleaching, laser intensity was adjusted for each field of view. Images were acquired on behaving mice that were head-fixed under the microscope on a custom-built running wheel. Time series data were acquired with a field of view of 525.17 $\mu m^2$ × 525.17 $\mu m^2$ (512 × 512 pixels) at 30 Hz for 90 s, 200 to 300 $\mu m$ below the pial surface in the retrosplenial (RSC) and secondary motor cortices (M2), respectively.

## Two-photon microscopy analysis

All raw fluorescence images were preprocessed with Suite2p [37], which included 2D image registration, cell detection, and time series extraction. All labeled cells were confirmed or removed based on visual inspection of both the image and time series. Briefly, we first imposed a strict skewness threshold of 2 and then manually discarded cells if the corresponding time series exhibited sudden jumps representative of significant z-displacements. For each recording, the time series of all cells was spatiotemporally decomposed using singular value decomposition. The first right singular vector, representing the time series of the first principal component, was smoothed with a 0.5-s sliding window and was correlated with the locomotion trace. The same was performed on the time series of each individual cell.

## Electrophysiology recordings

The 32-channel silicon probes (Cambridge NeuroTech, type H10b) with a 32-channel headstage (Intan technologies, C3324) were connected via a SPI interface cable (Intan technologies, C3216) to an USB interface board (Intan technologies, C3100). Data was recorded at 20 kHz using RHX software. Mice were anesthetized using the same FMM cocktail as previously described and head-fixed in a holding tube. The head of the mouse was aligned in all 3 axes to the coordinate system of the probe manipulator. Small perforations in the COMBO window film above the midbrain (AP: 4.00 mm, ML: 1.00) were created using a drill (0.5 mm diameter), and the probe was lowered into the midbrain at a speed of 2 $\mu m/s$. Data was acquired at 3 different depths (DV: −2.20 mm, −2.60 mm, and −3.60 mm below the brain surface) in both hemispheres during separate recordings. Concurrent optogenetic stimulation and electrophysiological recordings were performed 10 min after reaching the desired depth. In cases where the separate recordings were performed on different days, the perforations in the window were sealed with Kwik-Cast silicone sealant. CM-Dil (Thermo Fisher, CellTracker) was applied to the probe prior to brain insertion to allow post hoc confirmation of the recording site.

## Electrophysiology analysis

Electrophysiological recordings were analyzed using kilosort3 [55], and spike sorting results were manually curated using the phy software (https://github.com/cortex-lab/phy). Only units with stable responses across the duration of the session and high kilosort quality scores were included in subsequent analysis. To obtain continuous firing rate estimates, single unit responses were binned with a bin size of 1 ms and convolved with a Gaussian kernel with a standard deviation of 10 ms. For visualization in Fig 5, the firing rate of each cell was smoothed with a 70 ms window, and then trial-averaged and z-scored to the 4.5 s before stimulus onset. The Gramm toolbox [56] was used to visualize raster plots in Fig 5. For cell response-type

classification in S8G Fig, the trial averaged continuous firing rate of each cell was correlated to different stimulus windows. Cells that were positively correlated ($r > 0.2$) with the stimulus timing were defined ON cells, cells that negatively correlated with the stimulus timing ($r < -0.2$) deactivated cells, and cells that correlated positively ($r > 0.2$) with a window from stimulus offset to 800 ms after stimulus offset were named OFF cells.

## Open-field foraging task

Control mice and those with a COMBO window were handled by the experimenter for at least 2 days prior to the open field. Mice were placed in a 40 cm × 40 cm square arena (Stoelting) with opaque gray walls. A single diffuse LED light source was placed above the arena to ensure uniform lighting, and food pellets were scattered throughout the arena to promote exploration. A 1.3 megapixel monochromatic camera (Flir, BFS-U3-13Y3M-C) was positioned above the arena to capture the full field and recorded the animals' behavior for 10 min each at 20 Hz. Females were tested before males and the arena was cleaned between each animal. Mouse activity tracking was performed in Matlab using open-source code provided by Zhang and colleagues [57] as well as custom scripts. Mouse position data was downsampled by a factor of 3 and smoothed using a 0.5-s window before calculating the distance and speed between time points. Tortuosity was calculated as described in Zong and colleagues [30]. Briefly, the tortuosity of a given window (1.2-s sliding window) was calculated as the ratio between the actual distance traveled by the mouse and the length of the straight path between 2 points. Then, the median tortuosity of all time points where the mouse was running (speed threshold defined as the average of the 75th percentile of running speeds in control animals) was calculated.

## Facial videography

Videographic recordings were acquired using a 1.3 megapixel monochromatic camera (Flir, BFS-U3-13Y3M-C) either in the dark (two-photon imaging, fUS + visual stimulation) or in the presence of a small LED light (facial expressions, fUS + optogenetic stimulation). Videos for the facial imaging and during optogenetics experiments were acquired at 20 Hz, while those during two-photon calcium recordings were acquired at 30 Hz. One camera was always positioned such that the frame contained the full face, from ear to snout, and in some cases also the front 2 paws. In some experiments, a second camera was positioned with a field of view that captured the entire body of the animal. In general, two or three 875 mm IR LED arrays (Kemo Electronic M120) were pointed at the face to enable recordings in the dark. A 720 nm (Hoya, R72 E49) filter was attached to the camera to remove non-IR light. Videographic recordings were initiated using a TTL trigger that was aligned to the start of two-photon or fUS imaging acquisition.

## Habituation to head fixation

Prior to a recording session, all animals were gradually habituated to a head-fixed context. During the first 1 to 2 days of handling by the experimenter, animals were allowed to explore the behavior rig. Over the next 5+ days, the animals were head-fixed with the duration of fixation increasing on each subsequent day starting from 10 min and eventually reaching 60 min. Additional experimental components such as a computer screen or water spout were gradually introduced over the course of the habituation procedure.

## Head-fixed behavioral analysis

During two-photon videographic recordings, the front paws were tracked using DeepLabCut [58]. The resulting x and y position time series were used to create a distance measurement at every time point that represented the movement of the paw(s) relative to the previous frame. A locomotion time series was then created by averaging the movement time series of the left and right paw, which was then smoothed with a 0.5-s (15 frame) sliding window. Videographic recordings during optogenetic stimulation were analyzed using FaceMap [59]. The motion energy was extracted from ROIs placed over the whisker pad of the animal and of the wheel (without any body part visible), to obtain a proxy of whisking and locomotion, respectively. Whisking and locomotion time series were z-scored on a trial-by-trial basis to the 40 s before stimulus onset. To define trials with and without optogenetically induced locomotion in S8 Fig, a threshold was applied to the average z-score during the period between stimulus onset and 10 s after stimulus offset ($>$2 with locomotion, $<$2 without locomotion).

## Facial expression analysis

Facial expression analysis was performed in accordance with the methods described in Dolensek and colleagues [25]. After being habituated to a running wheel and water spout, animals were each administered 5 trials (2 s, 120 s inter-stimulus interval) of 20% sucrose and 10 mM quinine in subsequent runs. Similarly, baseline periods were defined as 2-s long intervals before any stimulus was presented and in which the animal exhibited no locomotion and minimal orofacial movement. HOG descriptors for each video frame were extracted with the following parameters: 32 pixels per cell, 1 cell per block, and 8 orientation bins.

Histogram of oriented gradient (HOG) facial analysis was performed in Matlab using custom scripts. Emotion prototypes were created for each mouse individually. Prior to HOG extraction, facial images were cropped to include only the face, ear, and whiskers of the mouse. An ROI containing the spout was also marked so that corresponding indices could be removed from all future HOG analyses. A neutral prototype was first created by averaging the HOG vectors from the previously described baseline periods. The HOG time series from stimulus runs were then correlated (Pearson's correlation coefficient) with this neutral prototype. The HOG vectors of the 10 least correlated time points during sucrose/quinine delivery were averaged to create the pleasure/disgust prototypes, respectively. To examine the neutral, pleasure, and disgust responses during different conditions, a time-resolved correlation coefficient was computed between each of the 3 emotion prototypes and the HOG vectors of the sucrose and quinine stimulus runs. This resulted in a neutral, pleasure, and disgust similarity time series for each of the sucrose and quinine runs, from which periods around stimulus/condition onset were extracted and visualized.

## Immunohistochemistry

A subset of animals ($N$ = 6 COMBO, $N$ = 5 control) were anesthetized with a ketamine (120 mg/kg) and xylazine (16 mg/kg) cocktail and transcardially perfused with 0.1 M phosphate-buffered saline (PBS), followed by 4% paraformaldehyde (PFA) in PBS. The brain was extracted and fixed in 4% PFA overnight at 4˚C, followed by submersion in 20% sucrose in 0.1 M PBS for up to 1 week at 4˚C.

The 50-μm thick coronal brain sections were acquired using a vibratome (Leica) and stored in cryoprotection solution at −20˚C for up to 1 week prior to staining. For GFAP staining, sections were transferred to 0.1 M PBS for 30 min, followed by washing in 0.1 M PBS containing 0.5% Triton X-100 for 10 min. Afterward, the sections were incubated in blocking solution, containing 10% blocking agent (Normal Serum Block, CAT# 927502, Biolegend) in 0.1 M PBS

with 0.5% Triton X-100 for 2 h. Subsequently, the sections were transferred to blocking solution containing 1:1,000 primary antibody rabbit Polyclonal anti-GFAP (PA5-16291, Thermo Fisher Scientific) for 24 h at 4˚C. Sections were washed 4 times in 0.1 M PBS with 0.5% Triton X-100 for 5 to 10 min each and then incubated in blocking solution containing 1:400 secondary antibody donkey Polyclonal anti-rabbit Alexa Fluor Plus 488 (A32790, Thermo Fisher Scientific) for 2.5 h. Sections were washed 4 more times in 0.1 M PBS with 0.5% Triton X-100 for 5 to 10 min and then mounted on glass slides using Fluoromount-G with DAPI (Thermo Fisher Scientific). Individual sections were imaged using a slide scanner (Olympus) using a 10× objective.

For virus expression and electrode track identification, brain sections were sliced, stored, and mounted as described above, but without any additional staining procedure. Individual sections were imaged using a slide scanner (Olympus) using a 4× objective.

### GFAP fluorescence quantification

To quantify and compare GFAP fluorescence throughout the area spanned by the window, we imaged 3 sections per brain corresponding to approximately bregma +2.0 mm AP, bregma −1.0 mm AP, and bregma −3.0 mm AP. Similar slices were imaged and quantified in both control animals and those with a COMBO window. GFAP immunofluorescence was then quantified using custom Matlab scripts. In each slice, 6 ROIs with a diameter of 250 μm were manually selected uniformly across the cortex, focusing on superficial layers and the pial surface. The GFAP fluorescence was taken as the average intensity within each ROI.

### Supporting information

**S1 File. COMBO_cup.stl.**
(STL)

**S2 File. COMBO_flat.stl.**
(STL)

**S3 File. COMBO_lateral.stl.**
(STL)

**S4 File. COMBO_posterior.stl.**
(STL)

**S5 File. COMBO_anterior.stl.**
(STL)

**S6 File. COMBO_Q1.stl.**
(STL)

**S7 File. COMBO_Q2.stl.**
(STL)

**S8 File. COMBO_Q3.stl.**
(STL)

**S9 File. COMBO_Q4.stl.**
(STL)

**S10 File. COMBO_bilateral.stl.**
(STL)

**S11 File. Head_plate.dwg.**
(DWG)

**S12 File. Head_plate_bilateral.dwg.**
(DWG)

**S13 File. Head_plate_holder.sldprt.**
(SLDPRT)

**S14 File. Head_plate_holder_top.dwg.**
(DWG)

**S15 File. Head_plate_holder_bottom.dwg.**
(DWG)

**S16 File. Brain_mold.stl.**
(STL)

**S1 Video. Facial videography during a trial of sucrose delivery.**
(MP4)

**S2 Video. Facial videography during a trial of quinine delivery.**
(MP4)

**S3 Video. Brain-wide fUS activity in awake mice in response to visual stimulation.**
(MP4)

**S4 Video. Facial videography and two-photon imaging in the retrosplenial cortex during locomotion.**
(MP4)

**S1 Fig. Head fixation part details.** (**a**) Computer-aided design of the standard COMBO window head plate (S11 File). The head plate attaches to the implant via 2 M1.4 through holes at the side and rear, as well as a peg in the front. Other custom head plate designs with the same features and relative distances can also be used for head fixation. (**b**) The standard head plate holder design (S13 File) consists of a single metal plate with the head plate outline cut halfway through the total thickness. Threaded M3 screws are welded into the head plate holes and grinded flush with the underside of the plate. M4 through holes allow for attachment to other commercial or custom parts for further stabilization. It is recommended that a machine shop helps with the fabrication of this part. (**c**) An alternative head plate holder design consists of a top and bottom plate (S14–S15 Files) that can each be laser cut and joined together with no custom fabrication. M3 screws can be used to secure the 2 layers together, and M4 screws to attach the head plate holder to other commercial or custom parts. Additional M3 screws can be attached via the underside of the holder using glue/epoxy.
(TIF)

**S2 Fig. The COMBO window is compatible with high-resolution MR receive coils.** (**a**) Whole-brain implant version ("bilateral") designed for magnetic resonance imaging (MRI) compatibility. Head fixation can be performed using ear bars in the scanner using this version. (**b**) Computer-aided design of a head plate compatible with the "bilateral" design that can be used outside of the scanner, if desired. (**c, d**) Schematics of the bilateral implant alongside a standard 1 cm loop coil (c) and a CryoProbe (d). As with any implant, air bubbles should be minimized when installing the implant and dental cement to avoid MR-related artifacts.
(TIF)

**S3 Fig. COMBO window assembly and installation instructions.** (**a**) Three-step diagram of the preparation of the COMBO window. Using the skull mold (S16 File) is recommended but not required for proper assembly. (**b**) Three-step diagram of the installation of the COMBO window after a cranial window has been created. The head plate can be installed at the same time as Steps 1 and 2 or at a later date. Detailed methods for both of the procedures are provided in **S1 Appendix**.
(TIF)

**S4 Fig. Sporadic GFAP fluorescence was observed in mice implanted with the COMBO window.** (**a**) Glial fibrillary acidic protein (GFAP) fluorescence in 2 example slices (top: bregma +2.0 mm AP, bottom: bregma −1.0 mm AP) of mice at 6 weeks after being implanted with the COMBO window. In both images, a localized increase of GFAP fluorescence can be seen in the left hemisphere. (**b**) Zoomed-in images of the elevated GFAP signal indicate that the immune response was found mostly in fibers located at or near the pial surface. Scale bars represent 500 μm.
(TIF)

**S5 Fig. Animal weight returns to pre-intervention range after COMBO window installation.** (**a, b**) The weight of individual animals implanted with a COMBO window ($N = 6$ mice) (a) and age-matched littermate controls ($N = 4$ mice) tracked for 10 days after installation (b). (**c**) Group-level weight values for animals with (blue) and without (black) a COMBO window installed. Points represent the mean ± SEM. Wilcoxon rank sum test, * $p < 0.05$, *** $p < 0.005$, uncorrected. Underlying data can be found in **S6 Data** and code in **S6 Code**.
(TIF)

**S6 Fig. Behavioral effects are consistent across sex and individuals.** (**a–c**) The total distance (a), speed (b), and tortuosity (c) of control and COMBO window mice separated into males ($n = 3$ mice) and females ($n = 4$ mice). Boxplots represent the median (center line), 25th and 75th percentiles (lower and upper box), and the 1st and 99th percentile (whiskers). Two-way ANOVA on ranks with main effects of sex and cranial window. Post hoc pairwise $t$ tests, Bonferroni corrected: n.s. $p > 0.05$. (**d**) Example prototypical disgust and pleasure facial expressions exhibited by 3 animals with a "cup" version of the COMBO window installed. Key features of the elicited disgust face include a flaring back of ear and an upturned snout, and of the elicited pleasure face include the forward movement of the ear and a downturned snout. Underlying data can be found in **S7 Data** and code in **S7 Code**.
(TIF)

**S7 Fig. The COMBO window enables chronic brain-wide acquisition of fUS data.** (**a, b**) Single-trial fUS traces from the SC plotted for each awake (**a**, $N = 4$ mice, $n = 46$ sessions) and anesthetized (**b**, $N = 5$ mice, $n = 32$ sessions) recording session. The arrows indicate the direction of the drifting grating visual stimulus. (**c**) fUS signal in the superior colliculi (SC) and primary visual cortex (VISp) of anesthetized mice covaries in response to drifting gratings in all 4 cardinal directions ($N = 5$ mice, $n = 32$ sessions). The dark black line and light gray shaded area represent the mean ± SEM, respectively, across sessions. (**d**) GLM results from 8 sessions of a single mouse recorded 7–9 weeks after being implanted with a COMBO window overlaid on the Allen Brain Atlas. Only voxels with an average T-score $> 2$ are displayed. Underlying data can be found in **S8 Data** and code in **S8 and S11 Codes**.
(TIF)

**S8 Fig. The COMBO window facilitates longitudinal experiments to interrogate neural circuits underlying behavior.** (**a**) Facial videography was used to monitor animal behavior on a

running wheel. Regions-of-interest (ROIs) placed over the whisker pad (violet) and wheel (green) were utilized to capture whisking activity and locomotion, respectively, induced by optogenetic stimulation of the secondary motor cortex (M2). (**b**) Consecutive trials from the same example session as in Fig 5C showing a robust and reliable increase in whisking in response to optogenetic activation of M2. Each trial was z-scored to a pre-stimulus baseline and rescaled between 0 and 1 for visualization purposes. (**c**) Region-wise segmented results of the optogenetically induced fUS activity. Regions are sorted by brain area, and only significantly modulated (correlation between stimulus timing and fUS signal) regions are included (significantly different from zero across sessions, $p < 0.01$, FDR-corrected). (**d**) Example coronal slices from the Allen Brain Connectivity Atlas (connectivity.brain-map.org/projection/experiment/287995889). AAV tracings after injection into the M2 (1) show widespread axonal projections from M2 to the striatum, the thalamus, and the midbrain (AP +2.30 mm to AP −3.90 mm). Scale bar represents 1 mm. (**e**) GLM analysis of fUS data in response to optogenetic stimulation of M2 ($N = 3$ mice, $n = 15$ sessions). In contrast to Fig 5E, here the trials were separated according to a strong or weak locomotor response (see Methods for threshold definition) to optogenetic stimulation. Only the voxels with T-scores significantly different from zero across sessions ($p < 0.05$, FDR-corrected) are shown. (**f**) Group-level fUS traces from the striatum (left) and thalamus (right) for trials with and without animal locomotion. The dark lines and lighter shaded areas represent the mean ± SEM, respectively, across sessions. (**g**) Pie chart showing the proportion of different cell response types observed in the midbrain (see Methods for cell response type definition). Underlying data can be found in **S9 Data** and code in **S9** and **S11 Codes**.
(TIF)

**S9 Fig. Blue light stimulation alone does not induce neural activation.** (**a**) Optogenetic activation of M2 results in widespread activation in ChR2-expressing mice. Average T-scores at indicated positions (distance to Bregma) are shown without thresholding (left, $N = 3$ mice, $n = 16$ sessions). Histograms of the same T-scores color-coded by significance ($p < 0.05$, FDR-corrected) (right). (**b, c**) Same as (a) for animals injected with an AAV9-CamkIIa-EGFP control virus in M2. Low T-scores and no significant voxels ($p < 0.05$, FDR-corrected) are observed for both the awake ($N = 3$ mice, $n = 12$ sessions) and anesthetized ($N = 3$ mice, $n = 14$ sessions) conditions. Underlying data can be found in **S10 Data** and code in **S10 Code**.
(TIF)

**S1 Table. Open field foraging task statistical analysis.**
(DOCX)

**S2 Table. Facial expression statistical analysis.**
(DOCX)

**S3 Table. Author contributions.**
(DOCX)

**S1 Appendix. COMBO window preparation and installation protocol.**
(DOCX)

**S1 Data. Underlying data for Fig 1F.**
(MAT)

**S2 Data. Underlying data for Fig 2C–2F, 2I and 2J.**
(MAT)

**S3 Data. Underlying data for Fig 3B, 3D and 3E.**
(MAT)

**S4 Data. Underlying data for Fig 4B, 4C, 4E and 4F.**
(MAT)

**S5 Data. Underlying data for Fig 5C, 5D, 5F and 5H–5J.**
(MAT)

**S6 Data. Underlying data for S5A–S5C Fig.**
(MAT)

**S7 Data. Underlying data for S6A–S6C Fig.**
(MAT)

**S8 Data. Underlying data for S7A–S7C Fig.**
(MAT)

**S9 Data. Underlying data for S8B, S8C and S8F Fig.**
(MAT)

**S10 Data. Underlying data for S9A–S9C Fig.**
(MAT)

**S1 Code. Code for Fig 1F.**
(M)

**S2 Code. Code for Fig 2C–2F and 2I–2J.**
(M)

**S3 Code. Code for Fig 3B, 3D and 3E.**
(M)

**S4 Code. Code for Fig 4B, 4C, 4E and 4F.**
(M)

**S5 Code. Code for Fig 5C, 5D, 5F and 5H–5J.**
(M)

**S6 Code. Code for S5A–S5C Fig.**
(M)

**S7 Code. Code for S6A–S6C Fig.**
(M)

**S8 Code. Code for S7A–S7C Fig.**
(M)

**S9 Code. Code for S8B, S8C, and S8F Fig.**
(M)

**S10 Code. Code for S9A–S9C Fig.**
(M)

**S11 Code. Helper functions.**
(ZIP)

## Acknowledgments

We thank Julia Kuhl for her help on figure illustrations and our colleagues from the Macé, Frank, and Gogolla labs for their comments on the manuscript.

## Author Contributions

**Conceptualization:** Bradley J. Edelman, Dominique Siegenthaler, Emilie Macé.

**Data curation:** Bradley J. Edelman, Dominique Siegenthaler, Paulina Wanken.

**Formal analysis:** Bradley J. Edelman, Dominique Siegenthaler.

**Funding acquisition:** Bradley J. Edelman, Dominique Siegenthaler, Nadine Gogolla, Thomas Frank, Emilie Macé.

**Investigation:** Bradley J. Edelman, Dominique Siegenthaler, Paulina Wanken, Bethan Jenkins, Bianca Schmid, Andrea Ressle.

**Methodology:** Bradley J. Edelman, Dominique Siegenthaler, Paulina Wanken, Bethan Jenkins, Nadine Gogolla, Thomas Frank, Emilie Macé.

**Project administration:** Bradley J. Edelman, Emilie Macé.

**Resources:** Nadine Gogolla, Thomas Frank, Emilie Macé.

**Software:** Bradley J. Edelman, Dominique Siegenthaler, Emilie Macé.

**Supervision:** Bradley J. Edelman, Nadine Gogolla, Thomas Frank, Emilie Macé.

**Validation:** Bradley J. Edelman, Dominique Siegenthaler, Paulina Wanken, Emilie Macé.

**Visualization:** Bradley J. Edelman, Dominique Siegenthaler, Emilie Macé.

**Writing – original draft:** Bradley J. Edelman, Dominique Siegenthaler, Emilie Macé.

**Writing – review & editing:** Bradley J. Edelman, Dominique Siegenthaler, Paulina Wanken, Thomas Frank, Emilie Macé.

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
