## [Editor Report · Decision Letter 0]

7 Nov 2023

Dear Dr Macé, 

Thank you for submitting your manuscript entitled "The COMBO window: A chronic cranial implant for multiscale circuit interrogation in mice" for consideration as a Methods and Resources Article by PLOS Biology. Please accept my apologies for the delay in getting back to you. 

Your manuscript has now been evaluated by the PLOS Biology editorial staff and I am writing to let you know that we would like to send your submission out for external peer review.

Once your full submission is complete, your paper will undergo a series of checks in preparation for peer review. After your manuscript has passed the checks it will be sent out for review. To provide the metadata for your submission, please Login to Editorial Manager (https://www.editorialmanager.com/pbiology) within two working days, i.e. by Nov 09 2023 11:59PM.

Kind regards,

Richard

Richard Hodge, PhD

rhodge@plos.org

PLOS

---

## [Decision Letter · Decision Letter 1]

10 Jan 2024

Dear Dr Macé,

Thank you for your patience while your manuscript "The COMBO window: A chronic cranial implant for multiscale circuit interrogation in mice" was peer-reviewed at PLOS Biology. It has now been evaluated by the PLOS Biology editors, an Academic Editor with relevant expertise, and by several independent reviewers. 

In light of the reviews, which you will find at the end of this email, we would like to invite you to revise the work to thoroughly address the reviewers' reports.

As you will see below, the reviewers find your work interesting and overall well done, but raise a few important points that need to be addressed.

Given the extent of revision needed, we cannot make a decision about publication until we have seen the revised manuscript and your response to the reviewers' comments. Your revised manuscript is likely to be sent for further evaluation by all or a subset of the reviewers.

**IMPORTANT - SUBMITTING YOUR REVISION**

*Re-submission Checklist*

*Published Peer Review*

*PLOS Data Policy*

Please note that as a condition of publication PLOS' data policy (http://journals.plos.org/plosbiology/s/data-availability) requires that you make available all data used to draw the conclusions arrived at in your manuscript. If you have not already done so, you must include any data used in your manuscript either in appropriate repositories, within the body of the manuscript, or as supporting information (N.B. this includes any numerical values that were used to generate graphs, histograms etc.). For an example see here: http://www.plosbiology.org/article/info:doi%2F10.1371%2Fjournal.pbio.1001908#s5

*Blot and Gel Data Policy*

Sincerely,

Christian Schnell (on behalf of Richard who is currently out of office)

Senior Editor

PLOS Biology

cschnell@plos.org

Richard Hodge, 

Senior Editor

PLOS Biology

rhodge@plos.org

REVIEWS:

Do you want your identity to be public for this peer review?

Reviewer #1: Yes: Noam Shemesh

Reviewer #2: Yes: Ravi Rungta

Reviewer #3: No

Reviewer #1: The manuscript by Edelman and colleagues (PBIOLOGY-D-23-02796R1) presents the "COMBO" window - a versatile solution for a chronic cranial window, which enables multimodal investigation of the mouse brain. The COMBO window is a designed for enabling access to most of the brain, with durability for long-term chronic investigations, and with flexibility to accommodate many different recording/manipulation modalities such as functional ultrasound (fUS), two-photon / optical imaging, electrophysiological recordings, and optogenetic manipulations. 

The authors first tested the durability and compatibility of their COMBO setup in terms of long-term inflammation, freely-moving behavior, and preservation of facial expressions. They then proceed to show proof-of-principle of the COMBO window's versatility by probing awake/anesthetized visual activity via fUS, 2P imaging of specific neurons during locomotion, optogenetic manipulation of pretargeted brain areas, and even electrophysiological recordings. In all cases, the authors were successful in clearly demonstrating both the durability/applicability of the setup. 

Overall, I find that this work is a tour-de-force of what's possible in terms of multimodal investigation of brain activity, and the COMBO window presented here represents the state-of-the-art in durable chronic implants for the mouse. I have no doubt that the freely available designs, including drawings and instructions, will represent a significant resource for the community and increase the reproducibility and quality of studies employing chronic implants. In addition, having such a resource can encourage others to engage/enter the field. The paper is also written exceptionally well and is easy to follow.

I only have a few minor comments:

1. One thing I haven't seen in this work is a more direct comparison against any other available implants (except perhaps for coverage issues): could the authors discuss advantages / disadvantages compared with other types of implants? 

2. While I highly appreciate that the authors tested preservation of behaviours and local inflammatory responses, would it make sense to look at more global metrics? E.g., corticosteroids in the blood, overall weight loss, reproduction, etc.?

3. In the Discussion, the authors fleetingly mention that compatibility with fMRI could be achieved. Could they expand the current drawings/instructions to include such a setting, at least in terms of the materials needed (without necessarily testing it in fMRI, just to facilitate the future investigations). Also, they should mention the complexity of integrating their COMBO setup with MRI reception coils.

4. The authors might want to mention that this setup could also benefit studies of resting-state activity and perhaps e.g. cite e.g. (Ma et al, Resting-state hemodynamics are spatiotemporally coupled to synchronized and symmetric neural activity in excitatory neurons, PNAS 2016) and/or similar papers. In fact, an MR compatible setup could greatly benefit the understanding of resting-state brain modes (e.g. Cabral et al Nature Communications 2023) via concurrent fUS or calcium imaging.

5. The COMBO window fits mice of the bl6 strain. Would modifications be needed for other widely-used strains? 

Reviewer #2: The article designs and characterizes a chronic cranial window implantation compatible for multiple techniques; optical and ultrasound imaging, fiber implantation (for optogenetics) and electrophysiology in mice. The design builds and extends nicely on previous work using PMP windows for combined ultrasound and optical imaging, and now provides a nice framework for incorporating fiber implantations and optogenetics. Furthermore, they show that their prep allows for monitoring of orofacial movements and behavior, with no noticeable impact of the implant. The data and methodology is nicely presented, and the experiments are well done and technically sound. Overall this is a nice article that should be published, I just have some minor comments which should be considered prior to acceptance.

Methods on P19: I would suggest adding a brief description on how the 1024 element probe is connected and operated with the 256 channel scanner (It is described in the reference, but I think should also be briefly described in the article).

Fig 3b, They show averaged traces across multiple mice (46 sessions), could they show what the traces look like from individual trials to give a sense of the feasibility of interpreting single trial recordings?

Movement and measurements of fUS signals. In fig 4b the authors show the displacement (in x and y) that occurs during locomotion. It would be informative to provide more information or discussion on the reliability of fUS signals during animal movement. 

The authors state that there are no large qualitative differences between locomotion and non-locomotion trials, however, it would be interesting to explore some of the differences and correlate them to the behaviour. This would also help illustrate the power of the approach. For example, you might expect the primary sensory cortex (hind/front paw regions), to show larger activation during locomotion due to the paws touching the wheel.

It would also be nice to show temporal quantiative traces of the fUS signals in a couple of brain regions for both the locomoting and non-locomoting conditions in the supplemental Fig. S6, similar to what is shown in 5f? 

P14: Compatibility of window with ephys. Since they need to drill through the window, are these terminal experiments? It would be important to describe this, as this represents a limitation worth discussing. What is the size of the "small perforations"?

A couple of control experiments for the light stimulation alone (in ChR2 negative mice) would be appreciated.

On this topic, light stimulation itself has been reported to cause local vasodilation and fUS signals (e.g. PMID 28139643). I realize that the authors do not image directly next to the fiber, where this artifact would be potentially expected to occur, but this could at least be briefly mentioned in discussion. This same paper should also be cited in the introduction in a different context, as it was the first study to combine fUS and 2P imaging with PMP windows, as expanded upon here.

Typo P16L31, visually visual-evoked

---

## [Editor Report · Decision Letter 2]

26 Apr 2024

Dear Dr Macé,

Thank you for your patience while we considered your revised manuscript "The COMBO window: A chronic cranial implant for multiscale circuit interrogation in mice" for publication as a Methods and Resources Article at PLOS Biology. This revised version of your manuscript has been evaluated by the PLOS Biology editors and the Academic Editor.

Based on our Academic Editor's assessment of your revision, I am pleased to say that we are likely to accept this manuscript for publication, provided you satisfactorily address the following data and other policy-related requests that I have provided below (A-E):

(A) In the animal ethics statement in the Methods section, I would be grateful if you could please specify that the mouse studies were specifically reviewed and approved by an IACUC or animal ethics committee. Please include the full name of the IACUC/ethics committee and include an approval number.

(B) You may be aware of the PLOS Data Policy, which requires that all data be made available without restriction: http://journals.plos.org/plosbiology/s/data-availability. For more information, please also see this editorial: http://dx.doi.org/10.1371/journal.pbio.1001797

-Supplementary files (e.g., excel). Please ensure that all data files are uploaded as 'Supporting Information' and are invariably referred to (in the manuscript, figure legends, and the Description field when uploading your files) using the following format verbatim: S1 Data, S2 Data, etc. Multiple panels of a single or even several figures can be included as multiple sheets in one excel file that is saved using exactly the following convention: S1_Data.xlsx (using an underscore).

-Deposition in a publicly available repository. Please also provide the accession code or a reviewer link so that we may view your data before publication. 

Figure 1F, 2C-F, 2I-J, 3B, 3D-E, 4B-C, 4E, 5C-D, 5F, 5H-J, S5A-C, S6A-C, S7A-C, S8B-C, S8F, S9A-C

(C) Please also ensure that each of the relevant figure legends in your manuscript include information on *WHERE THE UNDERLYING DATA CAN BE FOUND*, and ensure your supplemental data file/s has a legend.

(D) Per journal policy, as the custom code that you have generated (page 24) to is important to support the conclusions of your manuscript, its deposition is required for acceptance. Please ensure that the code is sufficiently well documented and reusable, and that your Data Statement in the Editorial Manager submission system accurately describes where your code can be found. 

(E) Please ensure that your Data Statement in the submission system accurately describes where your data can be found and is in final format, as it will be published as written there. 

We expect to receive your revised manuscript within two weeks. 

*Published Peer Review History*

*Press*

Sincerely,

Richard

Richard Hodge, PhD

rhodge@plos.org

PLOS

---

## [Editor Report · Decision Letter 3]

3 May 2024

Dear Dr Macé,

On behalf of my colleagues and the Academic Editor, Patrick Drew, I am pleased to say that we can accept your manuscript for publication, provided you address any remaining formatting and reporting issues. These will be detailed in an email you should receive within 2-3 business days from our colleagues in the journal operations team; no action is required from you until then. Please note that we will not be able to formally accept your manuscript and schedule it for publication until you have completed any requested changes.

PRESS

Kind regards,

Richard 

Richard Hodge, PhD

rhodge@plos.org

PLOS
